# In-silico calculation of soil pH by SCEPTER v1.0

Yoshiki Kanzaki[1], Isabella Chiaravalloti[2,3], Shuang Zhang[4], Noah J. Planavsky[2,3], Christopher T. Reinhard[1]

[1]School of Earth and Atmospheric Sciences, Georgia Institute of Technology, Atlanta, GA 30332, USA
[2]Department of Earth and Planetary Sciences, Yale University, New Haven, CT 06511, USA
[3]Yale Center for Natural Carbon Capture (YCNCC), Yale University, New Haven, CT 06511, USA
[4]Department of Oceanography, Texas A&M University, College Station, TX 77843, USA

*Correspondence to*: Yoshiki Kanzaki (ykanzaki3@gatech.edu), Christopher T. Reinhard (chris.reinhard@eas.gatech.edu)

**Abstract.** One of the soil properties most commonly measured to describe agronomic and biogeochemical conditions of soils is "soil pH". Soil pH measures the concentration of exchangeable $H^+$ that resides in bulk soil samples using extractants in the laboratory and thus differs from "porewater pH", which we define here as an in-situ measure of porewater $H^+$ concentration in soil/weathering profiles. The difference between the two pH measurements is often not fully known for a given system but could lead to a misunderstanding of soil conditions if the two measurements are directly compared. Agricultural soils are one of the targeted loci for application of the "Enhanced Weathering" (EW), a technique aimed at counteracting increasing anthropogenic carbon dioxide from burning fossil fuels. An increase in pH is thought to be one of key advantages of EW, given that the process can mitigate soil acidification and increase crop yields. As a result, fully evaluating the biogeochemical and agronomic consequences of EW approaches requires accurate simulation of both soil pH ($pH_s$) and porewater pH ($pH_{pw}$). This paper presents an updated version of the reactive transport code SCEPTER (Soil Cycles of Elements simulator for Predicting TERrestrial regulation of greenhouse gases), which enables simulation of bulk soil pH measurement in the laboratory in addition to porewater pH as measured in the field along with a more comprehensive representation of cation exchange with solid-phase constituents of bulk soil. We first describe the implementation of cation exchange in the SCEPTER model, then introduce conceptual modelling frameworks enabling the calculation of bulk $pH_s$. The validity of the model is examined through comparison of model results with soil pH measurements from mesocosm experiments of maize production with crushed basalt amendments. Finally, illustrative example simulations are shown demonstrating that a difference between $pH_s$ and $pH_{pw}$ can lead to significantly different estimates of soil alkalinization and carbon capture by EW for a given targeted pH in cropland systems.

# 1 Introduction

Continuous harvesting and excess use of nitrogen fertilizers commonly leads to acidification of agricultural soils, which may lead to soil degradation and food insecurity over the coming century (Kopittke et al., 2019). The addition of alkalinity to soils — traditionally through liming, the application of ground, relatively soluble (mostly carbonate) rock/mineral powder to soils (e.g., McLean, 1983; Thomas Sims, 1996; Rengel, 2003; Goulding, 2016) — is a widely utilized remedy to manage soil pH and stabilize crop yields. Addition of alkalinity to soil (including agricultural liming) has recently attracted attention because it can also sequester atmospheric $CO_2$ (e.g., Hamilton et al., 2007; Swoboda et al., 2022), an action that is urgently needed to help meet the climate targets delineated by the Intergovernmental Panel on Climate Change (IPCC, 2006, 2018). Indeed, Enhanced Weathering (EW) — the application of finely ground carbonate/silicate rock powder to soils — is one of a number of suggested schemes for actively removing anthropogenic $CO_2$ from the atmosphere at scale (e.g., Rau et al., 2007; Köhler et al., 2010; Taylor et al., 2016; Beerling et al., 2020; Vakilifard et al., 2021; Swoboda et al., 2022; Zhang et al., 2022; Kanzaki et al., 2023). In particular, applying basalt rock powder onto croplands/hinterlands has been suggested to be one of the most scalable, safe, and economically promising $CO_2$ removal schemes given the relatively low toxicity in basalt leachates, sustainable availability of basalt rocks, and a range of potential co-benefits (e.g., Strefler et al., 2018; Beerling et al., 2020; Goll et al., 2021).

The pH change induced by addition of basalt powder is central to the EW scheme because the resultant pH (reflecting e.g., soil buffer capacity, local climate and particle size distribution of the milled rock) must be optimal for crop growth (e.g., Fernández and Hoeft, 2009), and the application rate of basalt feedstock and resultant carbon capture will thus scale with the magnitude of desired pH increase (e.g., Kelland et al., 2020; Kantzas et al., 2022; Zhang et al., 2022; Dietzen and Rosing, 2023). However, interpretation of pH in soil is not always straightforward, because two different types of pH measurement may potentially be regarded as a pH reference for evaluating soil acidity. One is referred to as "soil pH" — defined here as $pH_s$ — which measures $H^+$ residing in bulk soil samples that is measured in the laboratory as the pH of liquid extractants (deionized water or $KCl/CaCl_2$ solution) of bulk soil samples taken from the field. The other is "porewater pH" — defined here as $pH_{pw}$ — which measures in-situ $H^+$ concentrations in porewater flowing through or remaining in the soil/weathering profiles (e.g., Geibe et al., 2006; Steiner et al., 2018). In agricultural/agronomic situations it is most common to measure $pH_s$ (e.g., Thomas, 1996), while models that simulate biogeochemical reactions and transport within soils, including dissolution of basalt during EW, typically calculate $pH_{pw}$ (e.g., Kelland et al., 2020; Kanzaki et al., 2022). Potential differences between these distinct tracers of soil acidity are poorly explored, and in many cases the heterogeneous continuum that exists between dissolved $H^+$ in pore fluids and exchangeable $H^+$ on soil cation exchange sites is not discussed (cf., Nielsen et al., 2017).

Here, we present a newly developed numerical scheme in an attempt to fill in this technical and knowledge gap and to develop a more mechanistic understanding of the difference between porewater pH and bulk soil pH. A numerical reactive

transport model — SCEPTER (Soil Cycles of Elements simulator for Predicting TERrestrial regulation of greenhouse gases; Kanzaki et al., 2022) — has been updated to enable simulations of soil pH ($pH_s$) along with porewater pH ($pH_{pw}$). We first present the essential updates to the SCEPTER code (Section 2.1) and then describe potential modelling frameworks for simulating soil pH with the updated version of the model (Section 2.2). Then, the validity of the model is examined through comparison between simulated and observed soil pH for a set of mesocosm experiments amending a natural soil/maize system with crushed basalt (Section 3). We then discuss the implications of the difference between porewater and soil pH for EW and the associated impacts on soil acidity by showing example simulations in which basalt feedstock is added to cropland soil using either $pH_s$ or $pH_{pw}$ as a target pH for EW deployment (Section 4). Finally, we provide a summary of conclusions and touch briefly on future directions for model development (Section 5).

## 2 Model description

The SCEPTER model simulates reactions and transport of solid, aqueous, and gas species within soil, including dissolution/precipitation of minerals, three-phase biogeochemical reaction, bio-mixing and uplift/erosion of solid phases, advective and diffusive transport of aqueous species, and gaseous diffusion (Kanzaki et al., 2022). The model is developed for simulating not only natural weathering processes, but also EW with its specific features that allow explicit bio-mixing of soil including tilling by farmers, addition of solid materials on the topsoil and tracking of particle size distributions which facilitates surface area calculation for individual solid species. This updated version of the code (v1.0) adds several new functions/options to the previously published version (v0.9; Kanzaki et al., 2022). Among them, implementation of cation exchange is essential to the simulation of soil pH as the uptake of cations by solid phase exchangers is a determinant factor of the exchangeable acidity and nutrient cycling in soils. We first describe the implementation of cation exchange in SCEPTER (Section 2.1) and then frameworks for simulation of soil pH using the current version of the code (Section 2.2). All symbols used in this study and their definitions are summarized in Table A1 in Appendix A.

### 2.1 Cation exchange in SCEPTER

The current version of SCEPTER allows cation exchange involving $H^+$, $Na^+$, $K^+$, $Ca^{2+}$, $Mg^{2+}$ and $Al^{3+}$ on any solid species specified by the user (on clay minerals and organic matter by default; Table 1). Cation exchange reactions are assumed to be in equilibrium, and their fundamental reactions can be written as reactions among surfaces of solid phase exchangers and the cations:

$$Z_\varsigma X(\theta)^- + \varsigma^{Z_\varsigma +} \Leftrightarrow \varsigma\text{-}X(\theta)_{Z_\varsigma} \qquad\qquad\text{, (1)}$$

where $Z_\varsigma$ is the valence number of cation $\varsigma$, $X(\theta)^-$ denotes exchangeable surface sites of solid phase exchanger $\theta$ and $\varsigma\text{-}X(\theta)_{Z_\varsigma}$ represents the cation $\varsigma$ adsorbed onto exchangeable sites of $\theta$. Eq. (1) should be regarded as a half reaction because the surface fraction of $X(\theta)^-$ must be very small compared to the surface sites where net local charge is zero because of adsorption under

natural conditions (Appelo, 1994). Physically relevant net cation exchange can then be written as a combination of Eq. (1) for a given cation and Eq. (1) for a reference/competing cation so that the combined reaction equation does not have $X(\theta)^-$. As a reference cation, $Na^+$ and $Cs^+$ have been considered (e.g., Appelo, 1994; Steefel et al., 2002; Steefel, 2009). Here, we use $H^+$ as a reference competing agent, with the net exchange reaction given as:

$$(1/Z_\varsigma)\varsigma^{Z_\varsigma+} + \text{H-X}(\theta) \Leftrightarrow (1/Z_\varsigma)\varsigma\text{-X}(\theta)_{Z_\varsigma} + H^+ \tag{2}$$

The equilibrium constant for Eq. (2) can be defined as follows:

$$K'_{\varsigma\backslash H,\theta} = \frac{f^{1/Z_\varsigma}(\varsigma\text{-X}(\theta)_{Z_\varsigma})[H^+]}{f(\text{H-X}(\theta))[\varsigma^{Z_\varsigma+}]^{1/Z_\varsigma}} \tag{3}$$

where $f(i)$ denotes the charge-equivalent fraction of surface species $i$, and $[j]$ represents the concentration of aqueous species $j$ (mol $L^{-1}$). The apparent equilibrium constant $K'_{\varsigma\backslash H,\theta}$ can vary as a result of surface fraction $X(\theta)^-$ and we adopt the formulation by Appelo (1994):

$$K'_{\varsigma\backslash H,\theta} = \eta_{H,\theta} K_{\varsigma\backslash H,\theta} \tag{4}$$

Here, $K_{\varsigma H,\theta}$ is the intrinsic equilibrium constant and $\eta_{H,\theta}$ is formulated as a function of $1-f(\text{H-X}(\theta))$ assuming that $f(X(\theta)^-)$ is proportional to $1-f(\text{H-X}(\theta))$ (Appelo, 1994):

$$\log \eta_{H,\theta} = -\alpha_\theta\{1 - f(\text{H-X}(\theta))\} \tag{5}$$

where $\alpha_\theta$ is assumed to be 3.4 by default.

The solution for the fraction of surface species can be obtained by considering mass balance at the exchangeable sites for each exchanger:

$$CEC_\theta = \sum_\varsigma Z_\varsigma \langle \varsigma\text{-X}(\theta)_{Z_\varsigma} \rangle \tag{6}$$

where $CEC_\theta$ is the cation exchange capacity of exchanger $\theta$ (eq $g^{-1}$) and $\langle i \rangle$ is the concentration of surface species $i$ (mol $g^{-1}$). By definition,

$$f(\varsigma\text{-X}(\theta)_{Z_\varsigma}) \equiv \frac{Z_\varsigma \langle \varsigma\text{-X}(\theta)_{Z_\varsigma} \rangle}{CEC_\theta} \tag{7}$$

Therefore, Eq. (6) can be alternatively written as

$$1 = \sum_\varsigma f(\varsigma\text{-X}(\theta)_{Z_\varsigma}) \tag{8}$$

Further, with Eqs. (3) and (4), Eq. (8) can be transformed into

$$1 = f(\text{H-X}(\theta)) + \sum_{\varsigma \neq H} \left( \frac{\eta_{H,\theta} K_{\varsigma\backslash H,\theta} f(\text{H-X}(\theta))}{[H^+]} \right)^{Z_\varsigma} [\varsigma^{Z_\varsigma+}] \tag{9}$$

Eq. (9) is solved for $f(\text{H-X}(\theta))$ once given a porewater chemistry and thermodynamic constants for exchange reactions (Table 2). Once $f(\text{H-X}(\theta))$ is obtained fractions of all surface species can be calculated using Eqs. (3)-(5).

In the previous version of SCEPTER, the key variables for tracking aqueous species are the total concentrations for individual dissolved elements (denoted as $c_\varsigma$ for dissolved element $\varsigma$). In the updated model, the tracked independent variables have been changed to the concentrations of free dissolved species (except for Si, for which $H_4SiO_4$ concentration is tracked), denoted as $c_\varsigma^1$. These $c_\varsigma$ and $c_\varsigma^1$ are related to one another by the following equation (Kanzaki et al., 2022):

$$c_\varsigma = c_\varsigma^1 + c_\varsigma^1 \sum_{i=2}^{n_\varsigma} K_{\varsigma,i} [\mathrm{H}^+]^{\gamma_{\varsigma,i,p}} \prod_{\varsigma' \neq \varsigma}^{n_{aq}} (c_{\varsigma'}^1)^{\gamma_{\varsigma,i,\varsigma'}} \prod_\varepsilon^{n_{gas}} (p_\varepsilon)^{\gamma_{\varsigma,i,\varepsilon}} \qquad , (10)$$

where the second term on the right-hand side is the sum of the concentrations of dissolved element $\varsigma$ other than $c_\varsigma^1$, denoted as the $i$-th species of dissolved element $\varsigma$ where $i \neq 1$, with $K_{\varsigma,i}$ being the thermodynamic constant for production of $i$-th aqueous species of dissolved element $\varsigma$, $\gamma_{\varsigma,i,p}$, $\gamma_{\varsigma,i,\varsigma'}$ and $\gamma_{\varsigma,i,\varepsilon}$ the stoichiometry of $\mathrm{H}^+$, dissolved element $\varsigma'$ and gas species $\varepsilon$, respectively, in the reaction that produces $i$-th aqueous species of $\varsigma$, $p_\varepsilon$ the partial pressure (atm) of gas species $\varepsilon$, and $n_{aq}$ and $n_{gas}$ the total numbers of independent aqueous and gas species, respectively (see Kanzaki et al. (2022) for more details). This modification of tracked independent variables (from $c_\varsigma$ to $c_\varsigma^1$) facilitates our implementation of cation exchange.

In accord with the implementation of cation exchange as well as modification of independent variables to track for aqueous species described above, the governing equation for aqueous species has been updated to:

$$\frac{\partial \phi \sigma \ell \beta_\varsigma^{aq} c_\varsigma^1}{\partial t} + \frac{\partial B_\varsigma^{ads} c_\varsigma^1}{\partial t} = -\frac{\partial \phi \sigma \ell v \beta_\varsigma^{aq} c_\varsigma^1}{\partial z} + \frac{\partial}{\partial z}\left(\phi \sigma \ell \tau_{aq} D_\varsigma \frac{\partial \beta_\varsigma^{aq} c_\varsigma^1}{\partial z}\right) + \sum_\theta^{n_{sld}} \gamma_{\theta,\varsigma} R_\theta + \sum_\kappa^{n_{xrxn}} \gamma_{\kappa,\varsigma} R_\kappa$$

$$+ \frac{\partial w B_\varsigma^{ads} c_\varsigma^1}{\partial z} - B_\varsigma^{ads} c_\varsigma^1 \int_0^{z_{ml}} E_\theta(z,z') dz' + \int_0^{z_{ml}} B_\varsigma^{ads} c_\varsigma^1(z') E_\theta(z',z) dz' \qquad . (11)$$

The first and second terms in the left-hand side of Eq. (11) denote the time rate of change of dissolved and adsorbed forms of $\varsigma$, respectively, with $\beta_\varsigma^{aq}$ and $B_\varsigma^{ads}$ ($m^{-3}$ L) defined as the factors to convert $c_\varsigma^1$ to $c_\varsigma$ and to the total concentration of element $\varsigma$ adsorbed onto solid phases, respectively. The first and second terms on the right-hand side show the advective and diffusive transport rates of dissolved forms of $\varsigma$, respectively, the third and fourth terms net supply of $\varsigma$ through dissolution/precipitation of solid phases and extra biogeochemical reactions, respectively, and the rest of the terms the advective transport (fifth term) and bio-mixing (sixth and seventh terms) rates of adsorbed forms of $\varsigma$ along with solid phase exchangers. The parameters to formulate the individual terms, reactions, and transport in Eq. (11) described above are tabulated in Table A1 in Appendix A. See Kanzaki et al. (2022) for further details on the reactions and transport schemes implemented in SCEPTER. The values of $\beta_\varsigma^{aq}$ and $B_\varsigma^{ads}$ can be calculated from Eqs. (10) and (3)-(9), respectively:

$$\beta_\varsigma^{aq} \equiv \frac{c_\varsigma}{c_\varsigma^1} = 1 + \sum_{i=2}^{n_\varsigma} K_{\varsigma,i} [\mathrm{H}^+]^{\gamma_{\varsigma,i,p}} \prod_{\varsigma' \neq \varsigma}^{n_{aq}} (c_{\varsigma'}^1)^{\gamma_{\varsigma,i,\varsigma'}} \prod_\varepsilon^{n_{gas}} (p_\varepsilon)^{\gamma_{\varsigma,i,\varepsilon}} \qquad , (12)$$

$$B_\varsigma^{\text{ads}} \equiv \begin{cases} \dfrac{1}{c_\varsigma^1} \sum\limits_\theta m_\theta M_\theta \langle \varsigma\text{-X}(\theta)_{Z_\varsigma} \rangle = \sum\limits_\theta \dfrac{m_\theta M_\theta CEC_\theta}{Z_\varsigma} \left( \dfrac{\eta_{\text{H},\theta} K_{\varsigma \backslash \text{H},\theta} f(\text{H-X}(\theta))}{[\text{H}^+]} \right)^{Z_\varsigma} & (\varsigma \in \{\text{Na, K, Ca, Mg, Al}\}) \\ 0 & (\text{else}) \end{cases} \quad , (13)$$

where $m_\theta$ and $M_\theta$ are the concentration (mol m$^{-3}$) and molar weight (g mol$^{-1}$) of solid species $\theta$, respectively.

The updated version of the governing equation for aqueous species (Eq. 11) is solved together with those for solid and gaseous species as described by Kanzaki et al. (2022), except that the calculation of surface speciation via cation exchange is additionally included during each update of porewater pH and aqueous speciation. Default capacities and thermodynamic constants of cation exchange are tabulated in Tables 1 and 2, respectively. Cation exchange can be switched on and off by specifying so in the switches.in input file. One can also modify the cation exchange parameters for any solid species using another input file cec.in; e.g., it is possible to assign different cation exchange parameters to different classes of organic matter that differ from the default values in Tables 1 and 2. Instructions for running example simulations from this paper are given in *Code Availability*.

### 2.2 Soil pH simulation by SCEPTER

In-silico calculation of bulk soil pH (pH$_s$) imitates the procedure to measure soil pH in the laboratory: sampling bulk soils, mixing them with an extractant solution (e.g., deionized water or KCl/CaCl$_2$ solution) at a given soil/solution ratio (e.g., 1:1 or 1:5 g/ml), bringing the mixtures to a short-term equilibrium, and measuring extractant solution pH (e.g., McLean, 1983; Thomas, 1996; Jones, 1999; Kissel and Sonon, 2008). Soil "buffer pH" — a measure of resistance of bulk soil to a pH change — can be calculated in silico using the same procedure but with a specified buffer solution (e.g., Thomas Sims, 1996; Sikora, 2006) instead of the extractant solutions implemented for measuring bulk agronomic pH. Our procedure for calculating soil (buffer) pH can be summarized as follows:

1. A "field simulation" is run, which can be fed by field observations.
2. Data from the field run are retrieved at a given model field depth and/or averaged over a given depth interval, including output for:
    a. Concentrations and cation exchange properties (e.g., Tables 1 and 2) of unextractable solid phases (e.g., silicates)
    b. Concentrations of exchangeable (i.e., dissolved plus adsorbed) cations and anions
    c. Concentrations of cations and anions in extractable solid phases (e.g., salts)
3. Boundary conditions for a "laboratory simulation" are determined based on Step 2 in order to realize a hypothetical laboratory "beaker/flask", where a bulk soil sample and an extractant solution (deionized water or electrolyte solution) are mixed homogeneously at a given soil/solution ratio.
    a. Concentrations of unextractable solid species obtained in Step 2 are given as the initial/boundary concentrations in an input file (parentrock.in) for the laboratory run. Those solid species are not

allowed to dissolve/precipitate in the laboratory run because of the short duration for soil pH measurements (e.g., Thomas, 1996), realized by setting their rate constants at zero in an input file (`kinspc.in`). Meanwhile cation exchange properties of the unextractable solid species are assumed to be the same as those in the field run (these can be specified in the corresponding input file `cec.in`).

b.  Exchangeable/extractable cations and anions are added to the calculation domain of laboratory "beaker/flask" as an appropriate combination of oxides and salts whose complete dissolution is allowed (Table 3). Note that dissolved inorganic carbon (DIC) is an exception and is instead added as the most labile class of organic matter (Table 3) so that carbon can be added without additional cations (compare e.g., carbonates). The amount of solid species added is calculated as $z_{lab}(1 - \phi_{lab})C_\varsigma M_\theta/\gamma_{\theta,\varsigma}$ (g m$^{-2}$) where $z_{lab}$ (m)

is the depth of the beaker/flask filled with the mixture of soil sample and solution, $\phi_{lab}$ is the volume ratio of fluid against solid phases calculated as $\phi_{lab} = \psi(\rho^{-1} + \psi)^{-1}$ with the soil/solution ratio used in the laboratory ($\psi$, g cm$^{-3}$) and bulk soil particle density ($\rho$, g cm$^{-3}$) observed in the in-silico field, $C_\varsigma$ is the concentration of exchangeable/extractable cation/anion $\varsigma$ (mol m$^{-3}$), $M_\theta$ is the molar weight of the added solid species $\theta$ (g mol$^{-1}$) and $\gamma_{\theta,\varsigma}$ is the mole of $\varsigma$ contained in 1 mole of $\theta$. When soil pH is measured in the mixture of bulk

soil sample and an electrolyte solution, corresponding salt is additionally added in the amount of $z_{lab}\phi_{lab}\ell c_\Theta M_\theta/\gamma_{\theta,\Theta}$ (g m$^{-2}$) where $c_\Theta$ and $\gamma_{\theta,\Theta}$ are the solution concentration (mol L$^{-1}$) of electrolyte $\Theta$ and mole of electrolyte $\Theta$ in 1 mole of salt $\theta$, respectively (e.g., $c_\Theta = 0.01$ mol L$^{-1}$ and $\gamma_{\theta,\Theta} = 1$ if $\theta = CaCl_2$ else $\gamma_{\theta,\Theta} = 0$ when $\Theta = CaCl_2$). When simulating soil buffer pH, the salt added corresponding to the electrolyte described above must be replaced by a series of solid phases corresponding to solute ingredients according to the recipe

of the buffer solution (e.g., Table 4 for a buffer solution by Sikora, 2006), enabling at the same time tracking of corresponding aqueous species with relevant aqueous diffusion coefficients and association/dissociation thermodynamics (e.g., Tables 5 and 6, respectively, for Sikora buffer solution). These constituents are added to the beaker/flask only once at the beginning of a laboratory simulation.

c.  The beaker/flask domain of the laboratory simulation is assumed to be a closed system for solid, aqueous,
and gaseous species, except for the addition of solid/salt phases at the beginning of the run described in Step 3b above, i.e., no advective transport for solid, aqueous and gaseous phases and no diffusive in- and out-fluxes of aqueous and gaseous species at the boundaries (specified in input files `frame.in` and `switches.in`).

4.  The laboratory simulation is run for long enough to achieve equilibrium.

Figure 1 schematically illustrates the procedure described above, from running a field simulation and sampling data from the in-silico field to soil pH measurement in the laboratory. As implied by the schematic (e.g., compare aqueous compositions illustrated for "porewater" in Step 2 and extractant solution in Step 4 of Fig. 1), porewater and soil pH can differ. Indeed, under

the conditions considered in our analysis a significant offset between $pH_s$ and $pH_{pw}$ is confirmed to be a general phenomenon, as discussed below. In the next section, we discuss the validity of our approach toward simulating $pH_s$ with SCEPTER using observed soil and porewater pH data from a mesocosm experiment along with other observed soil chemical characteristics. After examining the validity of the model (Section 3) we present examples of the model application to EW and discuss how the difference between porewater and soil pH can potentially lead to significant differences in the prediction of the amount of basalt feedstock required to achieve a given agronomic target pH in agricultural soils (Section 4).

## 3 Model validation

Before we examine the validity of our framework for soil pH calculation, the model's capacity to simulate cation exchange is compared with that of PHREEQC v3.0 (last access, 7 June 2023; Fig. 2). First, a series of experiments (Fig. 2a) is conducted with the two models with common cation exchange thermodynamics (Table 7) in order to compare the relationships between solution pH and base saturation at equilibrium when a solution with fixed concentrations of cations (1mM Na and 0.2 mM K) and different concentrations of nitrate (1 to 15 mM) is brought to equilibrium with a 1.1 meq $L^{-1}$ cation exchanger. Second, we perform a cation exchange simulation (Fig. 2b) in which a cation exchanging soil column initially homogeneously equilibrated with porewater consisting of 1 mM Na, 0.2 mM K and 1.2 mM $NO_3$ is flushed by 0.6 mM $CaCl_2$ solution through advection and dispersion with a Peclet number of 40, as in Appelo (1994), again using the same cation exchange properties (Table 7). We find negligible differences in the equilibria and dynamics of solutes and exchangeable cations between the two models (Fig. 2), indicating that the capacity of the current version of SCEPTER to simulate cation exchange is comparable to that of PHREEQC v3.0 (see also Supplementary Information for the effect of cation exchange in some of the previous example simulations run by Kanzaki et al. (2022)).

In order to validate our approach toward calculating bulk agronomic soil pH in the reaction-transport model, we compare a series of soil pH simulations fed by field simulation with observed boundary conditions to results from a mesocosm experiment. The mesocosm has been monitored since July 2022 at a greenhouse controlled under average growing season conditions. The field simulation is constrained from detailed measurements conducted in August 2022 (Table 8) as boundary conditions (Table 9). The field simulation is simplified as much as possible as the focus of this paper is simulation of soil pH (see Kanzaki et al., 2022, for some additional examples of field simulations fitted to observations; Supplementary Information). A detailed description of the mesocosm setup can be found in Chiaravalloti et al. (2023). Its tracked solid species are limited to soil organic matter and a "bulk" solid-phase species (i.e., a hypothetical species representing the solid phases other than soil OM dumped together as a whole) treated as two cation exchangers; tracked aqueous species include base cations (Na, K, Ca, Mg), $NO_3$ and Cl; and $CO_2$ gas. The tracked solid species (i.e., soil OM and "bulk" species) are assumed to have the same values for thermodynamic parameters for cation exchange except that they have different CEC values (120 and 3.176 cmol $kg^{-1}$, respectively) with their average constrained from the observed bulk soil CEC (8.9 cmol $kg^{-1}$). Measured porewater composition at 15 cm depth is used as the upper boundary condition for aqueous base cations in the field simulation so that

simulated porewater composition at 15 cm depth is consistent with observations (Fig. 3a). Aqueous $NO_3$ is added as $NH_4NO_3$ fertilizer at the upper boundary in the field simulation at the same rate of total N supply as the Urea-$NH_4$-$NO_3$ fertilizer applied to the mesocosm (24.210 gN m$^{-2}$ yr$^{-1}$). Upper aqueous Cl concentration takes a fitted value (Table 9) so that the simulated porewater pH at 15 cm depth is consistent with observations (Fig. 4a). Soil OM input is fixed at the value (Table 9) with which
simulated average OM concentration over the top 15 cm is consistent with observations (4.9 wt%). See Table 9 for more details on the boundary conditions for the field simulation.

Soil samples from the mesocosm experiments were homogenized from the top 15 cm of soil (dried at 60 °C overnight and sieved at 2 mm), and measured soil pH values and electrical conductivity values were obtained from a series of solutions: in deionized water at soil/solution ratios of 1:5, 1:2, 1:1 and 1:0.5 (g/cm$^3$); and in 0.0025, 0.005 and 0.01 M CaCl$_2$ solution at
1:1 soil/solution ratio (g/cm$^3$). The pH for each soil/solution slurry was measured with a Thermo Scientific Orion ROSS Ultra pH/ATC Triode paired with a Thermo Scientific Orion STARA2215 Orion Star A221 Portable pH Meter (ThermoFisher Scientific, Massachusetts). Electrical conductivity was measured by placing a few drops of the liquid from the soil/solution slurry on a HOBO U24 Conductivity Logger (U24-002-C) (Onset Computer Corporation, Massachusetts). We also measured buffer pH from a soil split using the method and recipe developed by Sikora (2006).

Soil pH simulations are conducted based on averaged data over top 15 cm of bulk soil from the field simulation described above, supplemented with the mesocosm observations according to the procedure described in Section 2.2. A series of soil pH values is calculated: in deionized water at soil/solution ratios of 1:5, 1:2, 1:1 and 1:0.5 (g/cm$^3$); and in 0.0025, 0.005 and 0.01 M CaCl$_2$ solution at 1:1 soil/solution ratio (g/cm$^3$) following Miller and Kissel (2010). We also calculate soil buffer pH where bulk soil over the upper 15 cm, deionized water, and Sikora buffer solution are mixed in 1:1:1 ratio (g:cm$^3$:cm$^3$) following the
recipe by Sikora (2006). The observation shows significant amounts of extractable $NO_3$ and Cl (Table 8), which probably exist as some forms of salts given measured electrical conductivities and are not explicitly simulated in the field run. Therefore, those extractable anions are added to the laboratory runs so that all major extractable/exchangeable elements measured in the mesocosm samples are consistent between the laboratory simulation and observations.

The simulated field run shows abundances of exchangeable cations over the top 15 cm that match well with observations
(Fig. 3b) with the optimized thermodynamic parameters for cation exchange (Table 9). Slight offsets might be attributable to chemical gradients developed especially for relatively strongly bound Ca and Mg, caused by CEC variation with depth. Simulated soil buffer pH is also consistent with observed buffer pH for the topsoil of the mesocosm (Fig. 4a). Although soil buffer pH was measured using Sikora buffer, we can confirm that the model can effectively reproduce the relationship between Sikora buffer pH and neutralized acid measured by the Sikora method (Fig. 4b). Therefore, in-silico measurement of soil buffer
pH should be directly comparable with the observational data. Simulated soil pH varies as a function of dilution by deionized water and/or the concentration of CaCl$_2$ in solution, a trend especially obvious when soil pH is plotted against electrical conductivity as shown in Fig. 5a (in-silico electrical conductivity is calculated from ionic strength assuming a conversion factor of 0.016 dS m$^{-1}$ mol$^{-1}$ L from Ponnamperuma et al., 1966; cf. Alva et al., 1991). This trend is also consistent with observations (Table 10 and Fig. 5a). The difference of soil pH in deionized water from that in 0.01 M CaCl$_2$ solution at the

same soil/solution ratio of 1:1 (g/cm$^3$), defined as $\Delta pH_{1:1}$ (Muller and Kissel, 2010), is also consistent with the mesocosm observations as well as the trend observed for U.S. soils by Miller and Kissel (2010) (Fig. 5b). Overall, with optimized thermodynamics of cation exchange the model can very closely reproduce observed porewater and soil (buffer) pH results for both our mesocosm experiments and previously published data (Miller and Kissel, 2010).

## 4 Example EW application

To illustrate the potential importance of distinguishing between $pH_s$ and $pH_{pw}$ and modelling both accurately, we present example simulations in which alkalinity addition to soils through EW for one year is limited by an assumed target pH and compare cases in which the target value is assumed to be $pH_s$ with an equivalent ensemble in which it is assumed to be $pH_{pw}$. Here, we consider another simple soil system which enables us to focus on any potential difference between $pH_s$ and $pH_{pw}$: tracked solid species include the "bulk" and SOM species; aqueous species are $Ca^{2+}$ and $NO_3$; and $CO_2$ is the only tracked gaseous species. Boundary conditions are those of an arbitrarily chosen field site from the midwestern U.S.A. (Table 11) and the cation exchange thermodynamics, soil respiration, and base saturation are correspondingly constrained from the observation at the site. More specifically, the system is tuned up with varying $Ca^{2+}$ concentration at the upper boundary and thermodynamic coefficients for Ca-H exchange and OM input to soil (3 unknowns) until the system satisfies the observed soil pH, exchangeable acidity and SOM wt% (6.058, 20.980 %CEC, 2.052 wt%, respectively; 3 observed knowns) at steady state. Mechanistically, the non-zero value of $Ca^{2+}$ concentration at the upper boundary can be taken to reflect the net result of historical liming at the site. We then add a "glassy basalt" solid species iteratively to meet a range of target pH values (6.2, 6.5 and 6.8) after one year of basalt application and use the model to estimate the rate of basalt application required to achieve a given target pH value. We run two ensembles, one with $pH_s$ as the operative target pH and one with $pH_{pw}$ as the target, allowing us to compare the estimated basalt feedstock application required to reach identical target pH when using $pH_s$ or $pH_{pw}$ as an index. For comparison with the target values, soil pH is calculated in a mixture of top 15 cm bulk soil and deionized water at 1:1 g/cm$^3$ ratio, while the average over top 15 cm is considered for porewater pH, calculated as $-\log(\int_0^{0.15}[H^+]dz/0.15)$. The observed data used for the initial spin/tune-up is from: Fick and Hijmans (2017) for temperature, Wang et al. (2019) for soil moisture, Reitz et al. (2017) for runoff, Poggio et al. (2021) for soil pH and OM, Walkinshaw et al. (2022) for cation exchange capacity, Pan et al. (2021) for nitrification rate, and ISRIC (2022) for base saturation. Basalt application simulations are all conducted as re-starts from the end of the same spin/tune-up described above, where glassy basalt is applied and mixed with bulk soil via tilling during initial 0.005 yr (~2 days). For glassy basalt, we use the kinetic law formulated by Brantley et al. (2008) and the thermodynamic calculation method used by Aradóttir et al. (2012) and Pollyea and Rimstidt (2017) and assumed a log normal distribution centered at 10 μm with 0.2 log unit standard deviation for the initial particle size distribution and the chemical composition in the caption of Table 11. See Table 11 for additional details on model boundary conditions.

Depending on the pH reference (i.e., either soil pH, $pH_s$, or porewater pH, $pH_{pw}$), required amount of basalt is significantly different at any of the target pH values examined here (Fig. 6), though all are within comparable ranges to previous

EW deployments (e.g., Swoboda et al., 2022). Comparison of soil and porewater pH (Figs. 7 and 8) shows that variation in soil pH is more limited compared to that of porewater pH because soil pH largely reflects exchangeable acidity, which can more effectively buffer input of alkalinity compared to acidity of porewater although the total exchangeable acidity is dependent on the cation exchange capacity and initial base saturation of soil. Porewater pH is lower than soil pH at relatively low alkalinity input (e.g., at earlier time after basalt deployment and/or at deep depths, Figs. 7 and 8), given that in-situ porewater pH reflects higher soil $pCO_2$ while soil pH has lower re-equilibrated $pCO_2$ from conserved DIC because of dilution by deionized water. With higher alkalinity input (e.g., at later time after basalt deployment and/or at shallower depths, Figs. 7 and 8), porewater pH is higher than soil pH because soil pH has a maximum value set by the cation exchange capacity at 100% base saturation. In general, using $pH_{pw}$ as the index target requires higher alkalinity input via basalt dissolution for a given target pH value because $pH_{pw}$ is lower than $pH_s$ in the background and it requires $pH_{pw}$ to reach higher values at shallower depths to compensate for the lower background $pH_{pw}$ at deeper depths. Though only meant to be illustrative, the example simulations shown here demonstrate the importance of distinguishing between soil and porewater pH in numerical frameworks for representing soil pH regulation. Our results indicate that care must be taken in reporting and validation of simulated pH values in reaction-transport models, particularly when comparing to analytical data for a bulk (multiphase) parameter such as soil pH.

## 5 Conclusions

We update the SCEPTER model (v1.0) to simulate the mechanics of cation exchange, and an associated, newly developed framework that enables calculation of soil pH in silico. By comparing to observational measurements from mesocosm experiments, we demonstrate that soil pH simulation in SCEPTER can accurately reproduce systematic variations in observed porewater pH, soil pH and soil buffer pH, so long as a field simulation can be validated by accessory soil chemistry. We also present example simulations which focus on the application of the model to estimation of required basalt for agricultural soils to reach different target pH values through EW. We observe significant differences in response to an alkalinity input via basalt dissolution between porewater and soil pH, with important implications for diagnosing agricultural soils with respect to an optimal basalt deployment rate/style through EW and managing crop yields. Future model developments include an extension of cation exchange to a more generalized suite of sorption reactions, e.g., implementation of anion (e.g., $PO_4$) adsorption onto oxides (e.g., van der Zee and van Riemsdijk, 1988; McGechan and Lewis, 2002) as well as nutrient uptake by plants, to comprehensively predict nutrient cycling and productivity in cropland soils in parallel with anthropogenic alkalinity modification and $CO_2$ removal through EW. Also, tracking of additional potential aqueous pH-affecting agents (e.g., dissolved organic matter, e.g., Nambu and Yonebayashi, 1999; Grybos et al., 2009) will widen the soil conditions to which the model can be applied to estimate shifts in porewater and soil pH as a result of EW. Further and wider use of the current/future code coupled with mesocosm/field observations of porewater pH and soil pH is expected to enhance our mechanistic understanding of the agronomic and climatic impacts of EW in croplands.

**Code availability**

The source codes of the model are available at GitHub (https://github.com/cdr-laboratory/SCEPTER) under the GNU General Public License v3.0. The specific version of the model used in this paper is tagged as "v1.0.1" and has been assigned a doi (https://doi.org/10.5281/zenodo.10805268). A readme file on the web provides the instructions for executing the simulations.

**Supplement**

The supplement related to this article is available online at: https://doi.org/xxxxx.

**Author contributions**

YK and CTR designed and implemented the model with significant contributions from the other authors. IC obtained mesocosm data. SZ and YK compiled and filtered soil data from the literature. All authors contributed to the experimental design and YK conducted the experiments and analysed the results. All authors contributed to the writing of the paper.

**Competing interests**

The authors declare no competing interests.

**Acknowledgements**

We are grateful to two anonymous reviewers for their useful comments and to Sandra Arndt for editorial handling.

**Appendix A: list of symbols**

Table A1 tabulates all symbols used in this study and their definitions.

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

Table 1. Default cation exchange capacity of solid species [a].

| ID | Name | $CEC_\theta$ [ceq kg$^{-1}$] | Ref./note [c] |
|---|---|---|---|
| ka | kaolinite | 16.2 | 1 |
| nabd, kbd, mgbd, cabd | Na-, K-, Mg-, Ca-beidellite | 70 | 2 |
| g1, g2, g3 | SOM [b] Class 1, 2, 3 | 330 | 2 |

[a] Those solid species that are not listed here are assumed to have zero cation exchange capacity.

[b] SOM — soil organic matter.

[c] (1) Beerling et al. (2020). (2) Parfitt et al. (1995).

Table 2. Default thermodynamic data of cation exchange [a].

| Reaction | $\log K_{\varsigma\backslash H}$ | Ref./note [b] |
|---|---|---|
| $Na^+ + H\text{-}X = Na\text{-}X + H^+$ | −5.9 | 1 |
| $K^+ + H\text{-}X = K\text{-}X + H^+$ | −4.8 | 2 |
| $Ca^{2+} + 2H\text{-}X = Ca\text{-}X_2 + 2H^+$ | −10.47 | 2 |
| $Mg^{2+} + 2H\text{-}X = Mg\text{-}X_2 + 2H^+$ | −10.786 | 2 |
| $Al^{3+} + 3H\text{-}X = Al\text{-}X_3 + 3H^+$ | −16.47 | 3 |

[a] The same set of thermodynamic data is assumed for any sold phase exchanger. Therefore, the notation of solid phase $\theta$ used in Section 2 is dropped in this table.

[b] (1) From modelled value at zero $f(H\text{-}X)$ in Appelo (1994). (2) Calculated from $\log K_{\varsigma\backslash Na} = 1.1, 0.507$, and $0.665$ for $\varsigma$ = K, Mg, and Ca, respectively, from Appelo (1994). (3) Calculated from $\log K_{Al\backslash Na} = 0.41$ from phreeqc.dat available in PHREEQC v.3 (Parkhurst and Appelo, 2013).

Table 3. Solid species to be dissolved in laboratory simulations [a].

| ID | Name | Composition | $M_\theta$ [g mol$^{-1}$] | $V_\theta$ [cm$^3$ mol$^{-1}$] | log $K_\theta^{\mathrm{ref}}$ [(mol L$^{-1}$)$^x$][b] | $\Delta H_\theta$ [kJ mol$^{-1}$] | Ref./ note [c] |
|---|---|---|---|---|---|---|---|
| na2o | Na$_2$O | Na$_2$O | 61.979 | 25.88 | 67.4269 | −351.636 | 1,2 |
| k2o | K$_2$O | K$_2$O | 94.195 | 40.38 | 84.0405 | −427.006 | 1,2 |
| mgo | MgO | MgO | 40.304 | 11.248 | 21.3354 | −150.139 | 1,2 |
| cao | CaO | CaO | 56.079 | 16.764 | 32.5761 | −193.832 | 1,2 |
| fe2o | FeO | FeO | 71.846 | 12 | 13.5318 | −106.052 | 1,2 |
| al2o3 | Corundum | Al$_2$O$_3$ | 101.962 | 25.575 | 18.3121 | −258.626 | 1,2 |
| sio2 | SiO$_2$ | SiO$_2$ | 60.085 | 22.688 | -2.71 | 13.97456 | 1,3 |
| caso4 | Anhydrite | CaSO$_4$ | 136.138 | 45.94 | −4.36 | −7.2 | 1,4 |
| nacl | Halite | NaCl | 58.443 | 27.015 | 1.5855 | 3.7405 | 1,2 |
| kcl | Sylvite | KCl | 74.551 | 37.524 | 0.8459 | 17.4347 | 1,2 |
| cacl2 | Hydrophilite | CaCl$_2$ | 110.986 | 50.75 | 11.7916 | −81.4545 | 1,2 |
| naoh | NaOH | NaOH | 39.9971 | 18.778 | - | - | 5 |
| amnt | NH$_4$NO$_3$ | NH$_4$NO$_3$ | 80.043 | 46.402 | - | - | 5 |
| g1 | SOM Class 1 | CH$_2$O | 30 | 20 | - | - | 5 |
| teas | Triethanolamine | C$_6$H$_{15}$NO$_3$ | 149.190 | 132.731 | - | - | 5 |
| ims | Imidazole | C$_3$H$_4$N$_2$ | 68.077 | 55.347 | - | - | 5 |
| mesmh | MES monohydrate | C$_6$H$_{13}$NO$_4$S•H$_2$O | 213.25 | 380.803 | - | - | 5 |
| gac | Acetic acid | CH$_3$COOH | 60.052 | 47.285 | - | - | 5 |

[a] Thermodynamic constants for solid species $\theta$ ($K_\theta$) are calculated as $K_\theta = K_\theta^{\mathrm{ref}}\exp(-\Delta H_\theta(T^{-1}-298^{-1})\mathcal{R}^{-1})$ where $T$ is temperature in K and $\mathcal{R}$ is the gas constant in units of kJ mol$^{-1}$ K$^{-1}$ ($\mathcal{R} = 8.314\times10^{-3}$ kJ mol$^{-1}$ K$^{-1}$). Solid species listed here are assumed to have decomposition rates that are represented by short turnover time (≤1 year) and do not depend on surface areas but their concentrations (see Kanzaki et al., 2022). Variation in kinetic constants does not affect the soil pH simulations as long as they are run long enough to attain equilibrium.

[b] Units change with $x$ depending on solid species.

[c] (1) $M_\theta$ and $V_\theta$ from Robie et al. (1978). (2) $K_\theta^{\mathrm{ref}}$ and $\Delta H_\theta$ from llnl.dat available in PHREEQC v.3 (Parkhurst and Appelo, 2013). (3) $K_\theta^{\mathrm{ref}}$ and $\Delta H_\theta$ are assumed to be the same as those for amorphous Si. (4) $K_\theta^{\mathrm{ref}}$ and $\Delta H_\theta$ from minteq.v4.dat available in PHREEQC v.3 (Parkhurst and Appelo, 2013). (5) Assumed to be undersaturated unconditionally. $M_\theta$ is calculated from chemical formula and $V_\theta$ is based on $M_\theta$ assuming density of 2.13, 1.725, 1.5, 1.124, 1.23, 0.56, and 1.27 g cm$^{-3}$ for NaOH, NH$_4$NO$_3$, SOM Class 1, triethanolamine, imidazole, MES (2-(N-morpholino)ethanesulfonic acid) monohydrate, and acetic acid, respectively.

Table 4. Sikora buffer composition [a].

| Solute | $c_\Theta$ [mol L$^{-1}$] |
|---|---|
| Triethanolamine | 0.0696 |
| Imidazole | 0.0137 |
| MES | 0.0314 |
| Acetic acid | 0.0893 |
| KCl | 2.00 |
| NaOH | 0.058 |

[a] From Sikora (2006) except that NaOH concentration is modified so that mixture of Sikora buffer with deionized water at 1:1 volume ratio has a pH of 7.5.

Table 5. Diffusion coefficients for aqueous species in Sikora buffer.

| Species | $a$ | $b$ | Ref./note [a] |
|---|---|---|---|
| Triethanolamine | 177.3 | - | 1,2 |
| Imidazole | 75.3 | - | 1,2 |
| MES | 380.803 | - | 1,3 |
| Acetate | 0.0251 | 21.57 | 4,5 |
| Cl | 0.0494 | 18.95 | 4,6 |

[a] (1) Diffusion coefficient (m$^2$ yr$^{-1}$) is calculated as $D = 0.4415(\mu_w^{-1.1}a^{0.6})^{-1}$ where $\mu_w$ is the water viscosity (mPa s) and $a$ is the molar volume of solute (cm$^3$ mol$^{-1}$) (Othmer and Thakar, 1953; La-Scalea et al., 2005). The water viscosity $\mu_w$ is calculated as $\mu_w = 0.024152\exp(4.7428(T - 139.86)^{-1}\mathcal{R}^{-1})$ where $\mathcal{R} = 8.314\times10^{-3}$ kJ mol$^{-1}$ K$^{-1}$ and $T$ is temperature in K, according to Likhachev (2003). (2) $a$ from La-Scalea et al. (2005). (3) $a$ is assumed to be equivalent to that of MES monohydrate. (4) Diffusion coefficient (m$^2$ yr$^{-1}$) is calculated as $D = a\times\exp(-b(T^{-1}-288^{-1})\mathcal{R}^{-1})$ where $a$ is the pre-exponential factor (m$^2$ yr$^{-1}$) and $b$ is the apparent activation energy (kJ mol$^{-1}$). (5) $a$ and $b$ from Schulz and Zabel (2006). (6) $a$ and $b$ from Li and Gregory (1974).

Table 6. Thermodynamic data for aqueous species in Sikora buffer [a].

| Reaction [b] | $\log K_{aq}^{ref}$ [(mol L$^{-1}$)$^x$] [c] | $\Delta H_{aq}$ [kJ mol$^{-1}$] | Ref./note [d] |
|---|---|---|---|
| TEA + H$^+$ = TEA(H)$^+$ | 8.09 | −33.6 | 1 |
| IM + H$^+$ = IM(H)$^+$ | 7.10 | −36.64 | 1 |
| MES(−H)$^-$ + H$^+$ = MES | 6.18 | −14.8 | 1 |
| AcO$^-$ + H$^+$ = AcOH | 4.48 | 0.41 | 1 |
| Cl$^-$ + H$^+$ = HCl | −0.67 | 0 | 2 |
| Cl$^-$ + Na$^+$ = NaCl | −0.777 | 5.21326 | 2 |
| Cl$^-$ + K$^+$ = KCl | −1.4946 | 14.1963 | 2 |
| Cl$^-$ + Mg$^{2+}$ = MgCl$^+$ | −0.1349 | −0.58576 | 2 |
| Cl$^-$ + Ca$^{2+}$ = CaCl$^+$ | −0.6956 | 2.02087 | 2 |
| Cl$^-$ + Fe$^{2+}$ = FeCl$^+$ | −0.1605 | 3.02503 | 2 |
| Cl$^-$ + Fe$^{3+}$ = FeCl$^{2+}$ | −0.8108 | 36.6421 | 2 |

[a] Thermodynamic constant ($K_{aq}$) is calculated as $K_{aq} = K_{aq}^{ref} \exp(-\Delta H_{aq}(T^{-1}-298^{-1})\mathcal{R}^{-1})$ where $\mathcal{R} = 8.314 \times 10^{-3}$ kJ mol$^{-1}$ K$^{-1}$ and $T$ is temperature in K.

[b] TEA — Triethanolamine; TEA(H)$^+$ — H$^+$-associated triethanolamine; IM — Imidazole; IM(H)$^+$ — H$^+$-associated imidazole; MES — 2-(N-morpholino)ethanesulfonic acid; MES(−H)$^-$ — H$^+$-dissociated MES; AcO$^-$ — Acetate anion; AcOH — Acetic acid.

[c] Units change with $x$ depending on reaction.

[d] (1) $K_{aq}^{ref}$ from Sikora (2006) and $\Delta H_{aq}$ from Goldberg et al. (2002). (2) From llnl.dat available in PHREEQC v3.0 (Parkhurst and Appelo, 2013).

Table 7. Boundary conditions for cation exchange simulations.

| Parameter [a] | Fig. 2a | Fig. 2b [b] |
|---|---|---|
| Solid species [c] | inrt | inrt |
| Aqueous species | Na, K, $NO_3$ | Na, K, Ca, $NO_3$, Cl |
| Gas species | - | - |
| OM [gC $m^{-2}$ $yr^{-1}$] | 0 | 0 |
| Nitrification [gN $m^{-2}$ $yr^{-1}$] | 0 | 0 |
| $J_\theta$ [g $m^{-2}$ $yr^{-1}$] | 0 | 0 |
| $N$ | 30 | 100 |
| $z_{tot}$ [m] | 0.5 | 0.5 |
| $w$ [mm $yr^{-1}$] | 0 | 0 |
| Bio-mixing | No | No |
| $\log r_H$ [m] | −5 | −5 |
| $q$ [m $yr^{-1}$] | 0 | 1 |
| $\sigma_0$ | 1 | 1 |
| $z_{sat}$ [m] | 1000 | 1000 |
| $CEC_{inrt}$ [ceq $kg^{-1}$] [d] | 0.04240 | 0.04240 |
| $c_{Na}^0$ [mmol $L^{-1}$] | 1 | 0 |
| $c_{K}^0$ [mmol $L^{-1}$] | 0.2 | 0 |
| $c_{NO_3}^0$ [mmol $L^{-1}$] | 1 to 15 | 0 |
| $c_{Ca}^0$ [mmol $L^{-1}$] | 0 | 0.6 |
| $c_{Cl}^0$ [mmol $L^{-1}$] | 0 | 1.2 |
| $\log K_{Na\backslash H,inrt}$ [e] | −1.0 | −1.0 |
| $\log K_{K\backslash H,inrt}$ [e] | −0.3 | −0.3 |
| $\log K_{Ca\backslash H,inrt}$ [e] | −0.4 | −0.4 |
| $\alpha_{inrt}$ | 0 | 0 |

[a] $J_\theta$ — addition rate of solid species $\theta$ at the upper boundary of the calculation domain, $N$ — number of grid cells in the calculation domain, $z_{tot}$ — total depth of the calculation domain, $w$ — uplift/erosion rate, $r_H$ — hydraulic radius of particles for solid phases, $q$ — annual runoff, $\sigma_0$ — water saturation ratio at the surface, $z_{sat}$ — water table depth, $CEC_{inrt}$ — cation exchange capacity assumed for "bulk" species, $c_\varsigma^0$ — concentration of $\varsigma$ at the surface ($\varsigma$ = Na, K, Ca, $NO_3$, Cl), $K_{\varsigma\backslash H,inrt}$ — thermodynamic constant for $\varsigma$-H exchange ($\varsigma$ = Na, K, Ca) at "bulk" solid species.

[b] Run as a restart from the spin-up whose boundary condition is given in the second column with $c_{NO_3}^0$ = 1.2 mmol $L^{-1}$ and $N$ = 100. To satisfy the Peclet number = 40, diffusion coefficients for all aqueous species are set at $6.5975\times10^{-2}$ ($m^2$ $yr^{-1}$) because tortuosity factor is calculated to be 0.3789 with assumed porosity and water saturation (Kanzaki et al., 2022).

[c] Only IDs of solid species are denoted. inrt — "bulk" species.

[d] Equivalent to 1.1 meq $L^{-1}$ with assumed porosity and water saturation.

[e] Calculated to be consistent with thermodynamic dataset Tipping_Hurley.dat available in PHREEQC v3.0 (Parkhurst and Appelo, 2013).

Table 8. Compositional data measured for mesocosm soil sample.

| Element | Porewater at 15 cm [mol L$^{-1}$] | Extractable/exchangeable [ppm] | Exchangeable fraction [%CEC] |
|---|---|---|---|
| Na | $9.5948 \times 10^{-5}$ | 13 | 0.6 |
| K | $7.1579 \times 10^{-4}$ | 57 | 1.6 |
| Mg | $1.9203 \times 10^{-4}$ | 179 | 16.8 |
| Ca | $1.3624 \times 10^{-3}$ | 996 | 56 |
| Al | $2.2872 \times 10^{-9}$ | - | - |
| NO$_3$-N | - | 120 | - |
| Cl | - | 1062 | - |

Table 9. Boundary conditions for mesocosm simulations.

| Parameter [a] | Field | Laboratory |
|---|---|---|
| Solid species [b] | inrt, amnt, g2 | inrt, amnt, g1, g2, cao, mgo, k2o, na2o, kcl, $(cacl2)^c$, (teas, ims, mesmh, gac, naoh)[d] |
| Aqueous species | Na, K, Ca, Mg, $NO_3$, Cl | Na, K, Ca, Mg, $NO_3$, Cl, (TEA, IM, MES, AcO)[d,e] |
| Gas species | $CO_2$ | $CO_2$ |
| OM [$gC\ m^{-2}\ yr^{-1}$] [f] | 1338 | 0 |
| $NH_4NO_3$ [$gN\ m^{-2}\ yr^{-1}$] | 69.172 | 0 |
| $J_\theta$ [$g\ m^{-2}\ yr^{-1}$] | 0 | Sections 2.2 and 3 |
| $N$ | 30 | 30 |
| $z_{tot}$ [m] | 0.5 | 0.05 |
| $w$ [$mm\ yr^{-1}$] | 1 | 0 |
| Bio-mixing ($z_{ml}$ [m]) | Fickian (0.25) | No |
| $\log r_H$ [m] | −5 | −5 |
| $q$ [$m\ yr^{-1}$] | 0.55 | 0 |
| $\sigma_0$ | 0.22 | 1 |
| $z_{sat}$ [m] | 1000 | 1000 |
| $CEC_{inrt}$ [$ceq\ kg^{-1}$] [f,g] | 3.176 | 3.176 |
| $CEC_{g2}$ [$ceq\ kg^{-1}$] [f,g] | 120 | 120 |
| $c^0_{Cl}$ [$mmol\ L^{-1}$] [f,h] | $2.68\times10^{-4}$ | - |
| $\log K_{Na\backslash H}$ [f,i] | −4.027 | −4.027 |
| $\log K_{K\backslash H}$ [f,i] | −4.474 | −4.474 |
| $\log K_{Ca\backslash H}$ [f,i] | −9.032 | −9.032 |
| $\log K_{Mg\backslash H}$ [f,i] | −8.704 | −8.704 |
| $\alpha$ [f,i] | 1.3 | 1.3 |

[a] $J_\theta$ — addition rate of solid species $\theta$ at the upper boundary of the calculation domain, $N$ — number of grid cells in the calculation domain, $z_{tot}$ — total depth of the calculation domain, $w$ — uplift/erosion rate, $z_{ml}$ — mixed layer depth, $r_H$ — hydraulic radius of particles for solid phases, $q$ — annual runoff, $\sigma_0$ — water saturation ratio at the surface, $z_{sat}$ — water table depth, $CEC_\theta$ — cation exchange capacity for solid species $\theta$, $c^0_{Cl}$ — concentration of Cl at the surface, $K_{\varsigma\backslash H}$ — intrinsic thermodynamic constant for $\varsigma$-H exchange ($\varsigma$ = Na, K, Mg, and Ca), $\alpha_\theta$ — coefficient to describe surface charge effect on cation exchange thermodynamics for solid species $\theta$ (Section 2.1; Appelo, 1994).

[b] Only IDs of solid species are denoted. Inrt — "bulk" species, amnt — $NH_4NO_3$, g1 — SOM Class 1 (most labile class), g2 — SOM Class 2 (second most labile class), na2o— $Na_2O$, k2o — $K_2O$, mgo — MgO, cao — CaO, kcl — KCl, cacl2 — $CaCl_2$, teas — Triethanolamine, ims — Imidazole, mesmh — MES (2-(N-morpholino)ethanesulfonic acid) monohydrate, gac — Acetic acid, naoh — NaOH.

[c] Added only when simulating soil pH in $CaCl_2$ solution.

[d] Added only when simulating soil buffer pH by Sikora (2006).

[e] Some of aqueous species in Sikora buffer are abbreviated. TEA — Triethanolamine, IM — Imidazole, MES — 2-(N-morpholino)ethanesulfonic acid, AcO — Acetate anion.

[f] Parameter values optimized to reproduce observation (Section 4).

[g] $CEC_\theta$ = 0 for solid species not listed here.

[h] See Section 3 for base cation concentrations at the upper boundary.

[i] Those values are applied only to bulk and SOM Class 2 species.

Table 10. Porewater and soil (buffer) pH of mesocosm.

| | Porewater pH at 15 cm | Soil pH in deionized water | | | | Soil pH in CaCl$_2$ | | | Buffer pH |
|---|---|---|---|---|---|---|---|---|---|
| | | 1:5 | 1:2 | 1:1 | 1:0.5 | 0.0025 | 0.005 | 0.01 | |
| Observation | 6.68 | 5.81 | 5.54 | 5.42 | 5.48 | 5.31 | 5.29 | 5.24 | 6.28 |
| Simulation | 6.68 | 5.74 | 5.52 | 5.36 | 5.20 | 5.32 | 5.29 | 5.25 | 6.27 |

Table 11. Boundary conditions for EW simulations.

| Parameter [a] | Field | Laboratory |
|---|---|---|
| Solid species [b] | inrt, amnt, g2, (gbas)[c] | inrt, amnt, g2, (gbas)[c], cao, mgo, k2o, na2o, g1 |
| Aqueous species | Na, K, Ca, Mg, $NO_3$ | Na, K, Ca, Mg, $NO_3$ |
| Gas species | $CO_2$ | $CO_2$ |
| OM [gC m$^{-2}$ yr$^{-1}$] [d] | 108.35 | 0 |
| Nitrification [gN m$^{-2}$ yr$^{-1}$] | 1.0059 | 0 |
| $J_\theta$ [g m$^{-2}$ yr$^{-1}$] | 0 (Depending on target pHs)[c,e] | Sections 2.2 |
| $N$ | 30 | 30 |
| $z_{tot}$ [m] | 0.5 | 0.05 |
| $w$ [mm yr$^{-1}$] | 1.013 | 0 |
| Bio-mixing ($z_{ml}$ [m]) [f] | Fickian (0.25) (Inversion (0.25))[c] | No |
| log $r_H$ [m] | −5 (PSD) | −5 |
| $q$ [m yr$^{-1}$] | 0.3514 | 0 |
| $\sigma_0$ | 0.2827 | 1 |
| $z_{sat}$ [m] | 1000 | 1000 |
| $CEC$ [ceq kg$^{-1}$] | 21.103 | 21.103 |
| $c_{Ca}^0$ [mmol L$^{-1}$] [d] | 0.1016 | 0 |
| log $K_{Ca\backslash H}$ [d,g] | −7.448 | −7.448 |

[a] $J_\theta$ — addition rate of solid species $\theta$ at the upper boundary of the calculation domain, $N$ — number of grid cells in the calculation domain, $z_{tot}$ — total depth of the calculation domain, $w$ — uplift/erosion rate, $z_{ml}$ — mixed layer depth, $r_H$ — hydraulic radius of particles for solid phases, $q$ — annual runoff, $\sigma_0$ — water saturation ratio at the surface, $z_{sat}$ — water table depth, $CEC$— cation exchange capacity assumed for "bulk" species and SOM, $c_{Ca}^0$ — concentration of Ca at the surface, $K_{Ca\backslash H}$ — thermodynamic constant for Ca-H exchange.

[b] Only IDs of solid species are denoted. inrt — "bulk" species, amnt — $NH_4NO_3$, g1 — SOM Class 1 (most labile class), g2 — SOM Class 2 (second most labile class), gbas — glassy basalt, na2o— $Na_2O$, k2o — $K_2O$, mgo — MgO, cao — CaO. Chemical composition of glassy basalt is given by the stoichiometry of $\gamma_{gbas,\varsigma}/\gamma_{gbas,Si}$ = 0.0809, 0.0084, 0.2439, 0.2722, 0.1251, 0.4683, and 1 for $\varsigma$ = Na, K, Ca, Mg, Fe, Al, and Si, respectively.

[c] Only enabled when basalt is applied in a field run or soil pH is simulated for basalt-applied soils.

[d] See Section 4 for the calculation of those parameter values.

[e] See Fig. 6.

[f] Bio-mixing is defined using a modified transition matrix ($K_{\theta,ij}$), which is a discretized form of continuous exchange function $E_\theta$ in Eq. (11) and can be formulated based on transport probability between soil layers $i$ and $j$ ($P_{\theta,ij}$). Inversion mixing in this paper is implemented as $K_{\theta,ij} = \delta z_i P_{inv}/\delta z_j$ if $i = j - 1$ or $i = j + 1$ or $i = n_{ml} + 1 - j$ else 0, where $P_{\theta,ij}$ is assumed to have a phase- and location-independent value $P_{inv}$ = 0.1 yr$^{-1}$, $\delta z_i$ is the thickness (m) of soil layer $i$ and $n_{ml}$ is the total number of mixed layers. See Kanzaki et al. (2022) for the formulation for Fickian mixing.

[g] Other thermodynamic constants for cation exchange are modified from their default values in Table 2 consistently with the change in $K_{Ca\backslash H}$, e.g., log $K_{\varsigma\backslash H}$ = −4.389, −3.289 and −7.764, for $\varsigma$ = Na, K and Mg, respectively.

Table A1. Symbols used in this study and their definitions.

| Symbol | Definition | Units |
|---|---|---|
| $B_\varsigma^{\text{ads}}$ | factor to convert $c_\varsigma^1$ to total concentration of element $\varsigma$ adsorbed onto solid phases | $\text{m}^{-3}\,\text{L}$ |
| $c_\Theta$ | solution concentration of electrolyte $\Theta$ | $\text{mol L}^{-1}$ |
| $c_\varsigma$ | total concentration of dissolved element $\varsigma$ | $\text{mol L}^{-1}$ |
| $c_\varsigma^1$ | concentration of free dissolved species for $\varsigma$ or $H_4SiO_4$ if $\varsigma = Si$ | $\text{mol L}^{-1}$ |
| $C_\varsigma$ | concentration of exchangeable/extractable cation/anion $\varsigma$ | $\text{mol m}^{-3}$ |
| $CEC_\theta$ | cation exchange capacity of exchanger $\theta$ | $\text{eq g}^{-1}$ |
| $D_\varsigma$ | diffusion coefficient of dissolved element $\varsigma$ | $\text{m}^2\,\text{yr}^{-1}$ |
| $E_\theta(z,z')$ | rate of particle transfer between locations at $z$ and $z'$ by bio-mixing | $\text{m}^{-1}\,\text{yr}^{-1}$ |
| $f(i)$ | charge-equivalent fraction of surface species $i$ | - |
| $\langle i \rangle$ | concentration of surface species $i$ | $\text{mol g}^{-1}$ |
| $[j]$ | concentration of aqueous species $j$ | $\text{mol L}^{-1}$ |
| $K_{\varsigma,i}$ | thermodynamic constant for production of $i$-th aqueous species of dissolved element $\varsigma$ | variable [a] |
| $K_{\varsigma\backslash H,\theta}$, $K'_{\varsigma\backslash H,\theta}$ | intrinsic and apparent equilibrium constants for cation exchange, respectively | $\text{mol}^{1-1/Z_\varsigma}\,\text{L}^{1/Z_\varsigma-1}$ |
| $\ell$ | unit conversion factor | $\text{L m}^{-3}$ |
| $m_\theta$ | concentration of solid species $\theta$ | $\text{mol m}^{-3}$ |
| $M_\theta$ | molar weight of solid species $\theta$ | $\text{g mol}^{-1}$ |
| $n_{\text{aq}}$, $n_{\text{gas}}$, $n_{\text{sld}}$ | total number of simulated aqueous, gaseous, solid species, respectively | - |
| $n_{\text{xrxn}}$ | total number of extra reactions | - |
| $p_\varepsilon$ | partial pressure of gas species $\varepsilon$ | atm |
| $R_\theta$ | net dissolution rate of solid species $\theta$ | $\text{mol m}^{-3}\,\text{yr}^{-1}$ |
| $R_\kappa$ | rate of $\kappa$-th extra reaction | $\text{mol m}^{-3}\,\text{yr}^{-1}$ |
| $t$ | time | yr |
| $X(\theta)^-$ | exchangeable surface sites of solid phase exchanger $\theta$ | - |
| $v$ | porewater advection rate | $\text{m yr}^{-1}$ |
| $w$ | advection rate of solid phases | $\text{m yr}^{-1}$ |
| $z$ | depth of weathering profile | m |
| $z_{\text{lab}}$ | depth of laboratory beaker/flask filled with the mixture of soil sample and solution | m |
| $z_{\text{ml}}$ | mixed layer depth | m |
| $Z_\varsigma$ | valence number of cation $\varsigma$ | - |
| $\alpha_\theta$ | factor to represent relation of $\eta_{H,\theta}$ to $f(X(\theta)^-)$ | - |
| $\beta_\varsigma^{\text{aq}}$ | factors to convert $c_\varsigma^1$ to total concentration of dissolved element $\varsigma$ | - |
| $\gamma_{\theta,\Theta}$ | mole of electrolyte $\Theta$ in 1 mole of solid species $\theta$ | - |
| $\gamma_{\theta,\varsigma}$ | mole amount of $\varsigma$ released upon dissolution of 1 mole of solid species $\theta$ | - |
| $\gamma_{\kappa,\varsigma}$ | stoichiometry of $\varsigma$ production in $\kappa$-th extra reaction | - |
| $\gamma_{\varsigma,i,\text{p}}$, $\gamma_{\varsigma,i,\varsigma'}$, $\gamma_{\varsigma,i,\varepsilon}$ | stoichiometry of $H^+$, dissolved element $\varsigma'$ and gas species $\varepsilon$, respectively, in production $i$-th aqueous species of $\varsigma$ | - |
| $\eta_{H,\theta}$ | factor to reflect the effect of surface potential on $K'_{\varsigma\backslash H,\theta}$ | - |

| | | |
|---|---|---|
| $\rho$ | bulk soil particle density | g cm$^{-3}$ |
| $\sigma$ | water saturation ratio | - |
| $\varsigma\text{-X}(\theta)_{Z_\varsigma}$ | cation $\varsigma$ adsorbed onto exchangeable sites of $\theta$ | - |
| $\tau_{aq}$ | tortuosity factor for solute diffusion in porewater | - |
| $\phi$ | porosity | - |
| $\phi_{lab}$ | volume ratio of fluid against solid phases | - |
| $\psi$ | soil/solution ratio used in the laboratory | g cm$^{-3}$ |

[a] Units can vary depending on considered reactions.

Step 1: Run simulation under field conditions

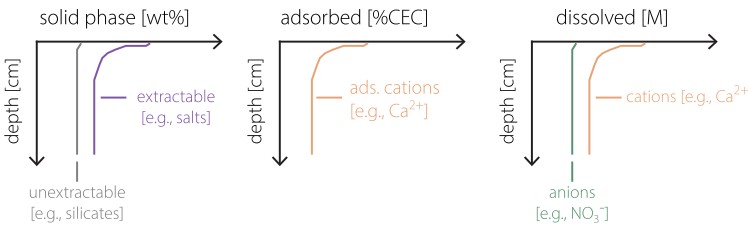

Step 2: In-silico sampling of field soil

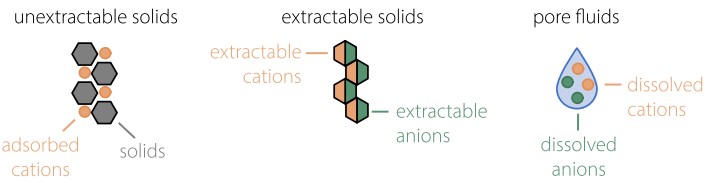

Step 3: In-silico processing of field soil

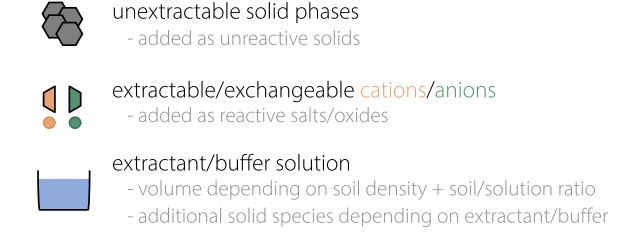

Step 4: Run laboratory simulation to equilibrium

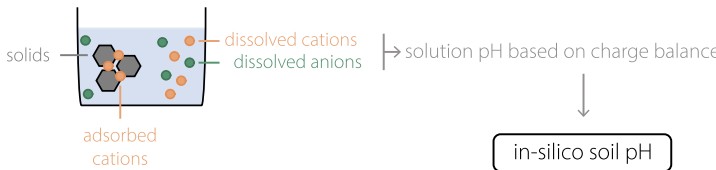

**Figure 1:** Schematic of soil pH calculation procedure. After a field simulation is run to represent a specific field soil (Step 1), in-silico field data are obtained (Step 2) for the concentrations of solid phases (left), adsorbed cations (middle) and dissolved cations and anions (right). In Step 3, sampled in-silico field data are converted to input data for a laboratory simulation in which extractable/exchangeable cations/anions are converted to a combination of salt/oxide phases to be added to the laboratory beaker/flask with additional phases depending on the extractant (or buffer) solution. In Step 4, these added phases are dissolved in the laboratory beaker/flask to reach equilibrium, after which the calculated solution pH corresponds to soil pH ($pH_s$) of the in-silico field soil in Steps 1 and 2.

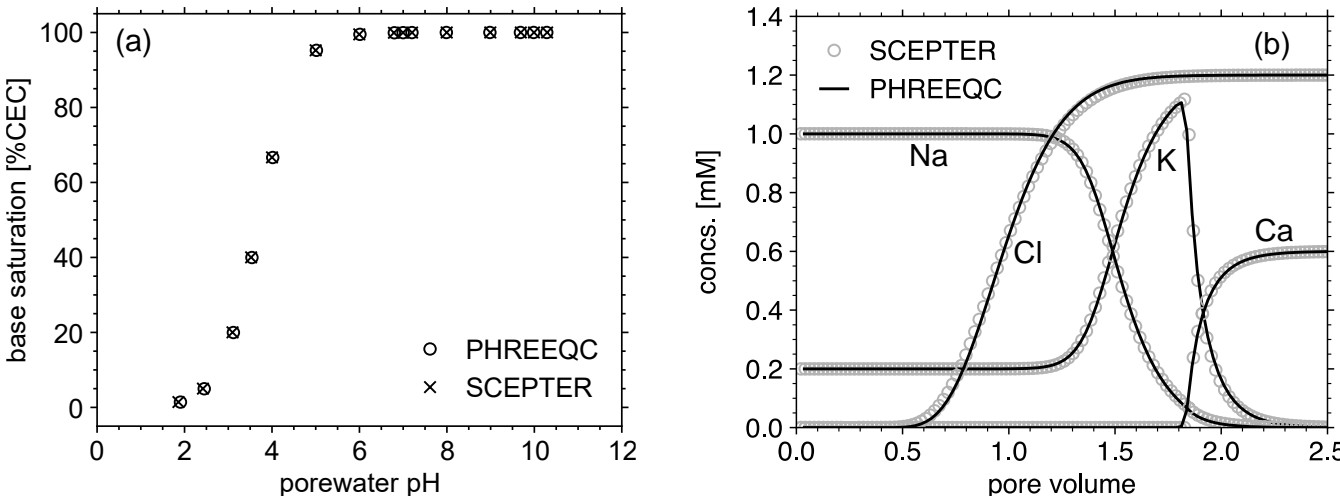

**Figure 2:** Comparison of cation exchange simulations between SCEPTER (v1.0) and PHREEQC (v3.0) with the thermodynamic dataset Tipping_Hurley.dat available in PHREEQC v3.0 (Parkhurst and Appelo, 2013). (a) Solution of 1 mM Na, 0.2 mM K and 1 to 15 mM $NO_3$ in equilibrium with a 1.1 meq $L^{-1}$ cation exchanger. (b) 0.6 mM $CaCl_2$ solution replacing a solution of 1 mM Na, 0.2 mM K and 1.2 mM $NO_3$ initially equilibrated with a 1.1 meq $L^{-1}$ cation exchanger homogeneously distributed along soil column through advection and dispersion with the Peclet number of 40 as in Appelo (1994). See Table 7 for the details on the experimental setups.

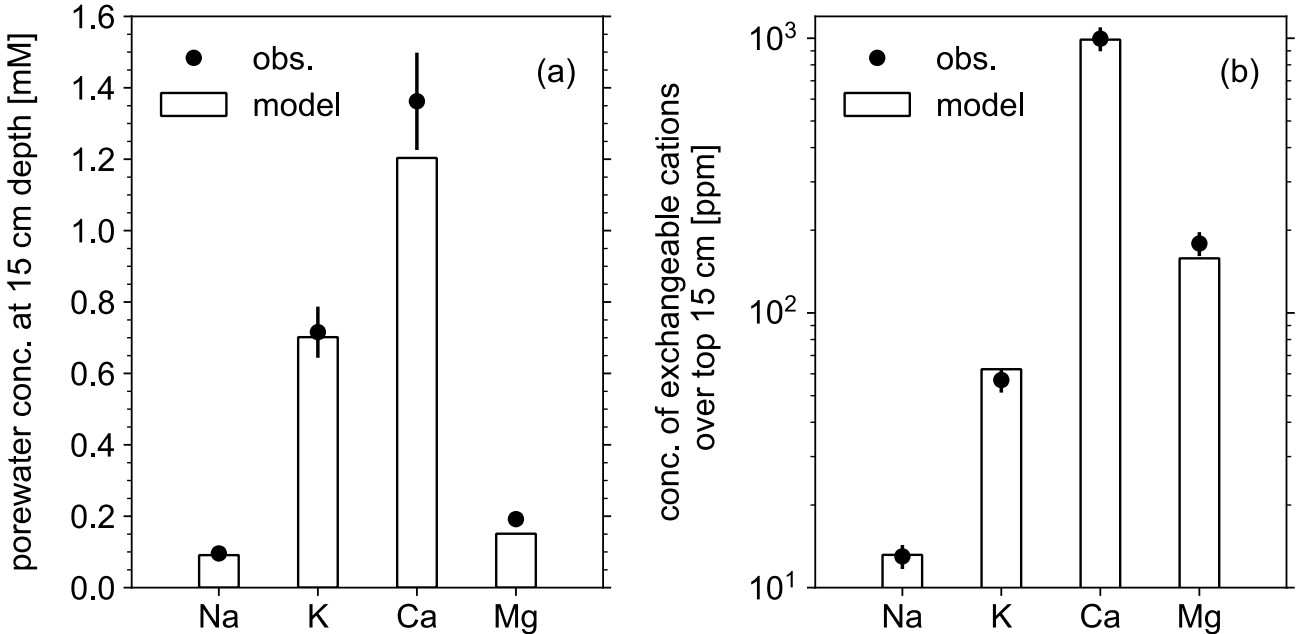

**Figure 3:** Comparison of soil composition between our model simulation and observations from the mesocosm experiments. (a) Porewater chemistry at 15 cm soil depth. (b) Concentrations of exchangeable cations over top 15 cm. A uniform 10% error is assumed for observational measurements (see Table 8).

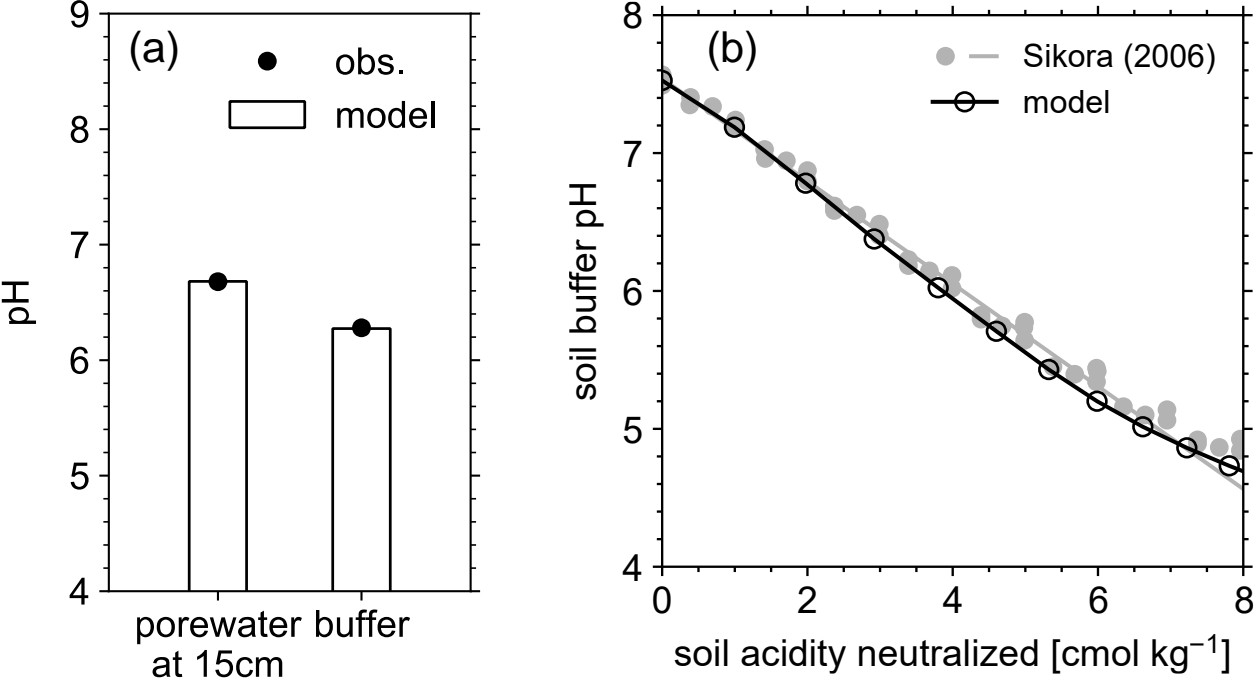

**Figure 4:** (a) Comparison of porewater and soil buffer pH between mesocosm observations and model simulation. (b) Data-model comparison of Sikora buffer pH (2006) as a function of neutralized acidity.

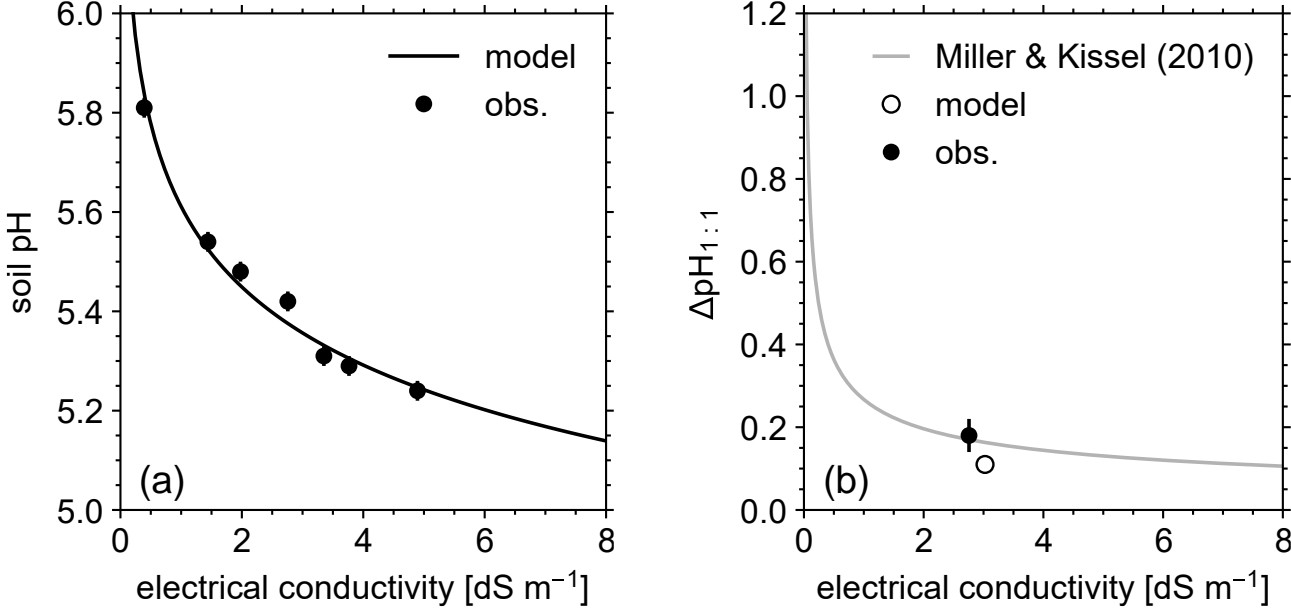

**Figure 5:** (a) Soil pH in deionized water at different soil/solution ratios and in $CaCl_2$ solution at different concentrations plotted against electrical conductivity for both simulations and mesocosm observations. (b) Difference in soil pH at 1:1 soil/solution $g/cm^3$ ratio between in deionized water and 0.01 M $CaCl_2$ solution ($\Delta pH_{1:1}$) plotted against electrical conductivity for both simulated and observed mesocosm, along with the $\Delta pH_{1:1}$ relationship with electrical conductivity derived for U.S. soils by Miller and Kissel (2010). In (a) and (b), measured pH is assumed to have a uniform error of 0.02.

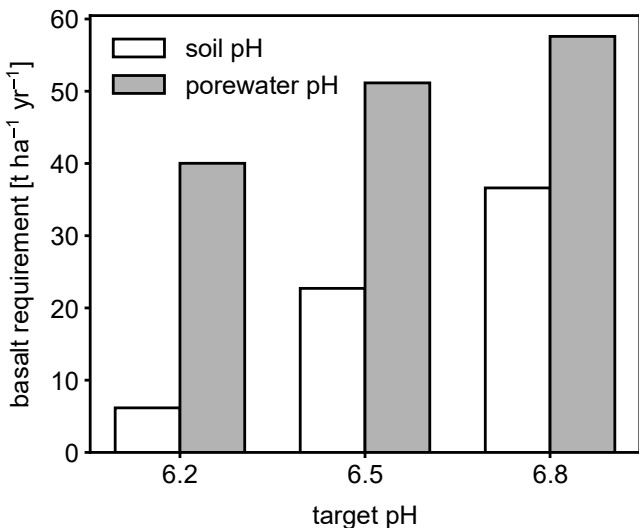

**Figure 6:** Basalt requirements for different target pH values after the first year following feedstock application using either bulk soil or porewater pH averaged over 0-15 cm as a pH reference value.

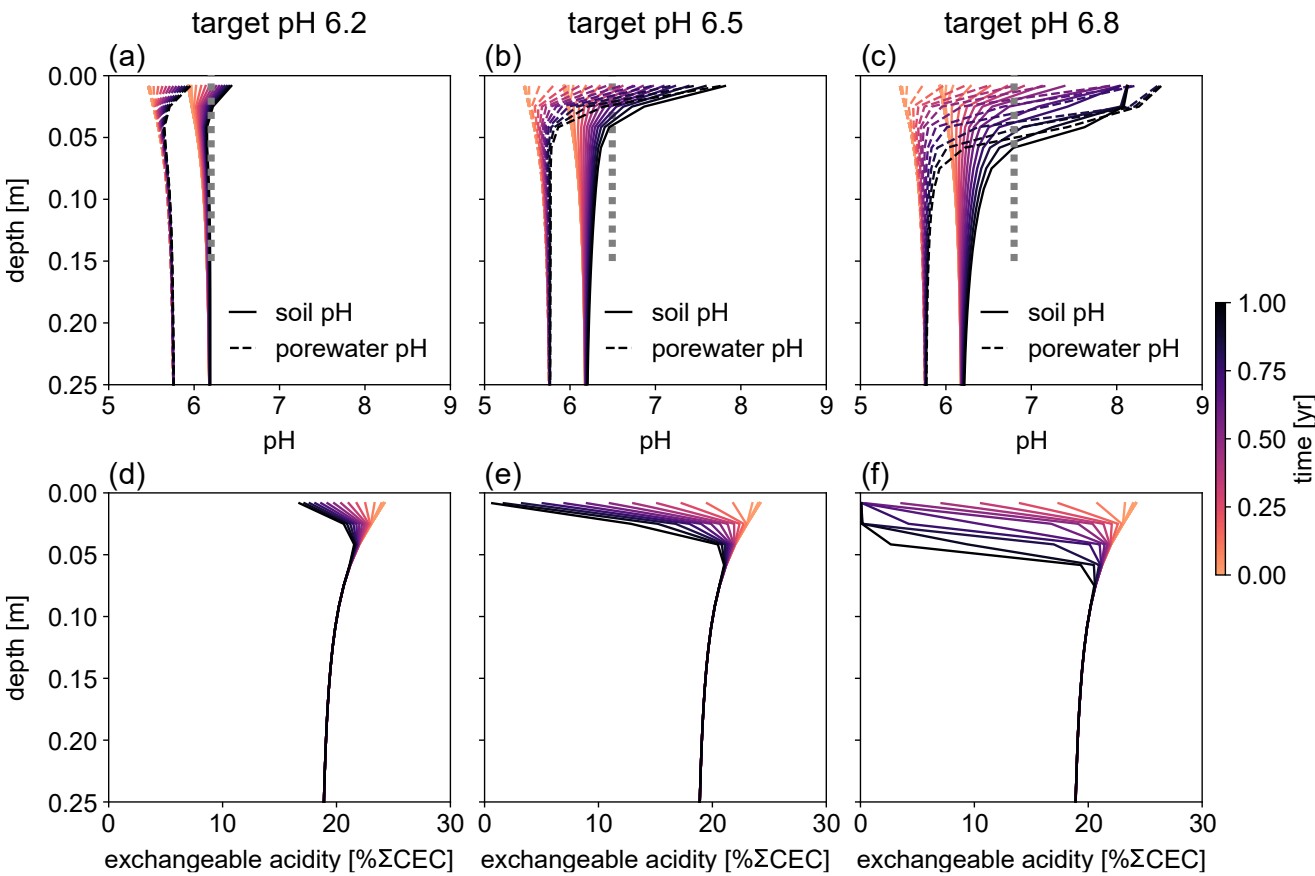

**Figure 7:** Evolution of soil and porewater pH (a-c) and exchangeable acidity (d-f) during the first year following basalt feedstock application at target pH values (vertical dashed lines) of 6.2 (a and d), 6.5 (b and e) and 6.8 (c and f) using soil pH averaged over 0-15 cm as a pH reference. Note that the curves of soil pH and porewater pH depicted here show values at individual depth points, while target pH values are integrated across the top 0-15 cm.

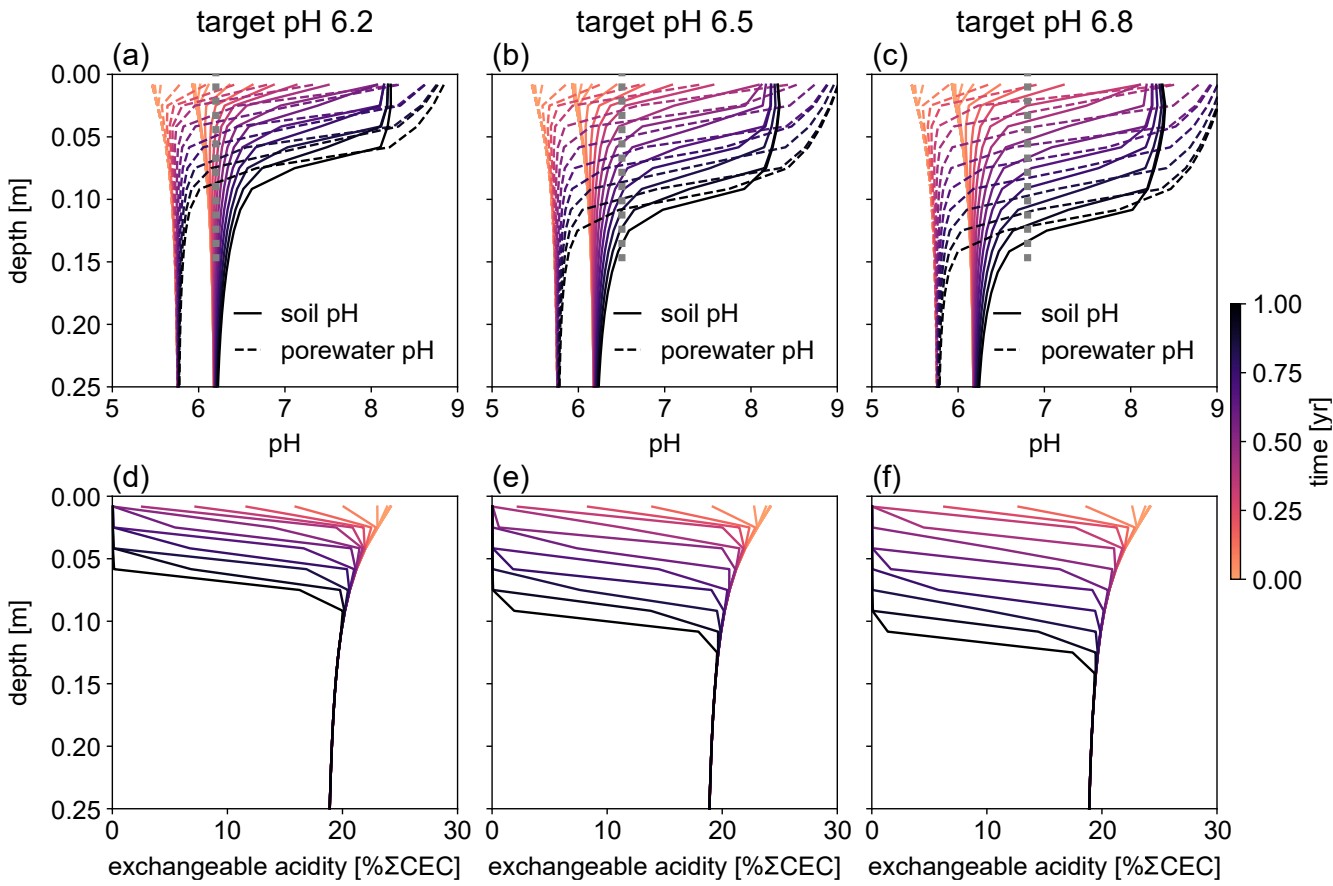

**Figure 8:** Evolution of soil and porewater pH (a-c) and exchangeable acidity (d-f) during the first year following basalt feedstock application at target pH values (vertical dotted lines) of 6.2 (a and d), 6.5 (b and e) and 6.8 (c and f) using porewater pH averaged over 0-15 cm as a pH reference. Note that the curves of soil pH and porewater pH depicted here show values at individual depth points, while target pH values are integrated across the top 0-15 cm.