# Peer review of "In-silico calculation of soil pH by SCEPTER v1.0"

_Geoscientific Model Development, 2023_

## Author Comment (AC1)

**Response to Referee #1**

We are grateful to Referee #1 for their very insightful and constructive comments. Our response to the reviewer's comments (original comments quoted in blue) and the corresponding revision are described in detail and separately below. The numbers of pages, lines, equations, tables and figures are those in the revised manuscript unless otherwise noted.

**Comment 1**: More emphasis on model development validation and not application; Better application examples to show importance of cation exchange on pH or carbon capture
"I find it difficult to evaluate this paper as it is not very clear to me, which conclusion the authors want to convey. In view of the journal, I presume that documenting the development of the model itself and not the application is of key relevance. In this case, a rigorous discussion of the results from the exemplary applications might not be expected. The applications would only serve the purpose to demonstrate that including ion exchange in the model can be of importance.", "Wouldn't it make it more sense to select one of the examples from the previous study (Kanzaki et al. 2022) and demonstrate that including ion exchange affects the predicted pH change (in the porewater) or the captured CO2 upon basalt amendment? Or to show, that cation exchange can exert influence on the saturation index and by this change mineralization rates. I find the story about the dependence of the required basalt dosage on how the pH is defined/determined a bit far-fetched.", "The consequences for basalt dosing as presented in Fig. 5 appear enormous as about eight times more basalt is required to increase porewater pH to 6.2 in comparison to the amount of basalt needed to increasing soil pH. However, this example is a bit misleading as the average pH for the top 15 cm after one year is used as a target. That is, the alkalinization front has only proceeded about 5 cm into the soil and the pore water pH in the deeper layers is already close to 6.2. In this case, the systematically lower pH in the soil pH has to be compensated by a higher pH in the upper 5 cm. I expect that the contrast is less when taking the average pH in the top 5 cm after 2 years as a target.", "Anyway, examples, which demonstrate that ion exchange is an important process when predicting the effect of enhanced weathering in terms of CO2 capture and resulting pH would be much more convincing. This would justify including this process in the model to achieve the model objectives. In conclusion the presented applications do not justify publication, and the justification should come from the model development itself."

Response:
Yes, the main goal of the manuscript is to document development of the model code itself, rather than exploring specific applications of the model, which fits with the purview of the journal. This is reflected in our discussion regarding the example application presented in the manuscript, which is relatively short (Section 4, the application section, occupying much less space than Sections 2 and 3, the sections describing model development/validation). We consider the simple application example presented in the manuscript to be appropriate, as it suggests different required amounts

of basalt depending on use of soil pH vs. porewater pH could lead to different porewater/soil pH profiles and different amounts of overall carbon removal; and further that these estimates are within the range that has been examined under the context of ERW (Swoboda et al., 2022). We have a study submitted elsewhere where application of the current code to croplands and the impacts of cation exchange are considered separately, consistent with the reviewer's suggestion that the focus of the current paper is on the model development/validation rather than application. In any case, we now include in the Supplementary Information a direct comparison to previous simulations with and without cation exchange.

Although we consider our simple application example suitable for demonstrating the potential importance of the differentiating porewater pH vs. soil pH, we fully agree with the reviewer that our results might be confusing without further clarifying the difference between pH over depth intervals vs. pH at individual depth horizons. We have attempted to clarify this in the revised text (see below).

Please see our response to Comment 2 by the reviewer regarding the issue of further validation of the model against PHREEQC.

Changes in manuscript (Page numbers/Line numbers):
We added more application examples such as those suggested by the reviewer to the Supplementary Information, in which we discuss the effects of adding cation exchange to previous simulations with an earlier version of the model code (P8/L221-222, Supplement).

We modified a relevant sentence to clarify that estimated amounts of basalt in Section 4 are consistent with the deployment range adopted for ERW studies summarized by Swoboda et al. (2022) (P10/L301-P11/L302).

We added more explanations to avoid confusion in Section 4 (P10/L290-291, P11/L311-312, P34-35), and added vertical dotted lines to Figs. 7 and 8 to show target pH values (P34-35).

**Comment 2**: Marginal extension of processes by including cation exchange; Non-straightforward procedures to calculate soil pH; Soil pH calculations with other programs; Benchmark with PHREEQC
"In view of model development, the presented progress is, in my opinion, only marginal. The extension of processes by including cation exchange is small compared to the complexity of the the previous version of the model.", "Calculating the pH in soil extracts is not straightforward in SCEPTER and requires some tricks including the evaporation of the porewater in silico and redissolving the obtained salts in the respective solutions and equilibrating the solution with the exchange complex. In particular, the addition of carbonate is cumbersome and occurs in the form of easily degradable organic matter. I did not understand why DIC could not be transferred in the form of highly soluble carbonates. In any case, the correctness of this procedure should be tested and documented, too. The model reproduced the measured soil pH of one soil taken from a mesocosm experiment but this does not proof that the whole procedure does not create artefacts.

The hassle of deforming the reactive transport model for aqueous equilibrium calculations implies, that it is unlikely that anybody will use the model for this purpose. That is because there a numerous alternatives for this purpose, or example PHREEQC, MINTEQ and other programs. Therefore, the authors should also verify the indirect model approach by comparing the predictions with those obtained from, for example, PHREEQC. The output of the model, including the composition of the exchange complex and the solution, could be directly used as an input in PHREEQC. The effect of diluting the porewater with CaCl2 solution can then be easily performed. In my opinion, comparing the predictions of SCEPTER and PHREEQC (or another chemical equilibrium model) as a benchmark should be included to demonstrate the correctness of the procedure to calculate soil pH."

Response:
We respectfully disagree with the reviewer that the extension of processes enabled by addition of cation exchange to the model is a marginal improvement, and we appreciate the opportunity to further clarify this point. Indeed, the potential number of applications to croplands can be significantly increased from that with the previously released code where neither cation exchange nor soil pH can be explicitly evaluated. This is because most soil data report only soil pH but not porewater pH especially for agronomic systems. Also, as described in lines 116-145, the governing equation for aqueous phase must be modified with the addition of cation exchange, and the master variables were changed as well, which was not trivial change in the code (e.g., file-size of the source code increased by ~17% from the previous release). In addition, multiple Python scripts were developed to enable soil pH calculation, and these are included in the v1.0 code release. Lastly, it is quite clear from the additional simulations now provided in the Supplementary Information (at the helpful suggestion of the reviewer) that the addition of these dynamics makes a first-order difference for predictions of carbon removal in some cases, depending on background soil characteristics.

The reason for adding DIC as "labile organic matter" rather than as solid carbonates is that adding carbonates is accompanied by addition of cations and alkalinity, which complicates the procedure to set up boundary conditions for laboratory runs. As long as the whole source of DIC (either labile organic matter or carbonate) is completely converted to DIC during the run and cation mass balance is satisfied (e.g., one has to make sure the amount of cations coming from carbonates and that from e.g., oxides sum up to the total amount of cations constrained from observations, if carbonate is used as DIC source), it does not numerically matter what "phase" is used for the DIC source in the code. As above, the choice of labile organic matter as a DIC source is simply for convenience — it allows one to avoid additional calculations regarding the amounts of cations added as e.g., oxides vs. carbonates to make sure that the total amount of cations added is consistent with field run results.

Regardless of the chosen program (e.g., even for thermodynamics-specific programs listed by the reviewer), soil pH calculation requires some postprocessing in order to be directly comparable to the procedure of actual soil pH measurements in the laboratory, as considered in

this study, given the non-straightforward re-distribution/re-equilibration of cations/anions among dissolved, exchangeable and extractable pools during the laboratory procedure. As far as we know, there are no such scripts nor even a framework publicly available for this comparison, thus motivating our attempt here to develop a framework and postprocessing tools to enable the calculation of soil pH in silico.

Nonetheless, we strongly agree with the reviewer that some benchmarking experiments with PHREEQC would be useful for readers, and we appreciate the suggestion. In the revised manuscript, we conduct benchmarking experiments on cation exchange for which PHREEQC scripts are available. We conduct 2 series of simulations to directly compare the capacity to simulate cation exchange between PHREEQC and SCEPTER: (1) one simulation in which a solution of a given composition is brought to equilibrium with an exchanger; and (2) one simulation in which transport (advection + dispersion) is added in addition to exchange equilibrium (Revised Fig. 2). The comparison suggests nearly identical cation exchange behavior for these simulations.

We fully agree with the reviewer that additional tests against more than one mesocosm experiment will be important for future work. However, the dataset utilized in this study is one of the few available that reports comprehensive porewater chemistry (including porewater pH) as well as soil (buffer) pH and exchangeable cations, enabling a detailed examination of the model's validity. We have added some caveats to this effect in the revised text.

Changes in manuscript (Page numbers/Line numbers):
We added/modified sentences to better describe importance in adding cation exchange to the model and enabling soil pH calculation (P2/L45, P8/L221-222, Supplement).

We added more explanations to better describe soil pH calculation (P7/L178-179).

We added a figure that compares PHREEQC vs. SCEPTER experiments with cation exchange (P8/L211-221, P21, P29).

We added additional text to emphasize that more comparisons with field observations are desirable in the future study (P11/L331-333).

**Comment 3**: Lacking pH-buffering from organic matter compounds with different functional groups or multiple pKa values
The third aspect is the adequateness of the approach to account for the buffer intensity in soils. In a first instance, concepts for describing cation exchange have been developed to describe the exchange of major cations between solution and constant charge surfaces. Here, the authors apply the concepts to account for the pH buffering in soils. This can cause some inconsistencies, in particular, when the contribution of organic matter to the buffer intensity is restricted to cation exchange. This approach implies, for example, that the pKa value of functional groups of organic matter depend on whether the background electrolyte includes Na or Ca ions. Furthermore, it restricts the description of acid/base properties of soil organic matter to only one acidity constant.

This might be sufficient in a limited pH range but, usually, multiple pKa models are used for an adequate description of the acid/base properties of soil organic matter. For the selected soil, the model preforms well but the versatility of the approach to account for the buffering intensity of other soils remains uncertain. However, I would not demand an extensive testing of the model to a large variety of soils but the possible limitations of the approach should be discussed.

Response:

This is a fair point, and we appreciate the reviewer raising it. The revised manuscript now includes a brief discussion of this point and emphasizes its importance as a topic for future research. Indeed, we would expand this point to include not just soil organic carbon but all of the individual species that contribute to the soil exchange complex (oxides, clays), which can also show a range of exchange characteristics. We do actually implement some major DOM species in the model; specifically, oxalic acid with two pKa values as well as its complexation with cations following Prapaipong et al. (1999, Geochim. Cosmochim. Acta, 63, 2547) and acetic acid with one pKa along with other dissolved organic compounds with one pKa values in Table 6. Therefore, we can use the current version of model if a DOM species is observed to contribute to porewater charge balance significantly and it has similar pKa value(s) to those already implemented in the model.

If the reviewer is also referring to a change in CEC (and exchange coefficients) at pH = pKa for a single solid organic phase because of functional group response to pH, we can address such a case with the current model by tracking two soil organic matter classes and assigning them correspondingly different CEC and thermodynamic constants while forcing them to have the same kinetic constants so that they behave as if they represent a single species. Similarly, tracking 3 classes of SOM with modified cation exchange coefficients and capacities can handle the situation where a single solid organic phase is characterized with two pKa's and 3 different CECs with pH.

Changes in manuscript (Page numbers/Line numbers):

We added a sentence in Conclusions that other pH buffering agents including DOM will be implemented in the future model development (P11/L329-P12/L331).

To avoid confusion, we added a description that each solid phase can have different thermodynamic constant for different cations and can be modified from the default values in Tables 1 and 2 (P6/L147-149).

**Comment 4**: Summary of Comments 1-3

"In conclusion, the paper presents a marginal extension of the processes included in SCEPTER. The presented validation is creative but does not represent a relevant application. The possible limitations of the approach are not discussed and the performance is only tested for one soil. I propose to accept the manuscript with major revisions and I suggest following improvements: 1) The validation should include benchmarking with aqueous equilibrium models, 2) the possible

limitations of using solely cation exchange to account for the pH buffering in soils should be rigorously discussed 3) the relevance of including cation exchange in the model should be demonstrated for the main application of the model, mineral weathering and CO2 sequestration. In general, the manuscript has a high quality, it is concisely written and the results are adequately presented figures and table. I have not tested the model and I hope other reviewers assessed the robustness and consistency of the model and investigated its usability and documentation."

Response:
We believe that we have sufficiently addressed the reviewer's concerns in our response to Comments 1-3 by the reviewer (see above), and we appreciate all of the points raised.

Changes in manuscript (Page numbers/Line numbers):
Please see our changes in manuscript in response to Comments 1-3 by the reviewer.

**Comment 5**: Figure 2
"Minor comments:", "Figure 2: Exchangeable fraction is in units of ppm. Is this correct? Are the other 99.99 % occupied by protons?"

Response:
Yes, we see how this could be confusing. The units of ppm here mean gram of exchangeable cations per a million gram of bulk soil rather than cation fraction of CEC or base saturation. We have revised the figure labeling to reflect this.

Changes in manuscript (Page numbers/Line numbers):
To avoid confusion, we changed the y-axis title of Figure 3 from "fraction" to "concentration". Please note that Figure 2 in the previous manuscript is now Figure 3 in the revised manuscript (P30).

---

## Author Comment (AC2)

**Response to Referee #2**

We are grateful to Referee #2 for their constructive and useful comments. Our response to the reviewer's comments (original comments quoted in blue) and the corresponding revision are described in detail and separately below. The numbers of pages, lines, equations, tables and figures are those in the revised manuscript unless otherwise noted.

**Comment 1**:
"Abstract: Generally fine, but there is an imbalance of overall description and results. The latter needs more emphasis or details. I suggest condensing the "general introduction" and expand the key findings and implications."

Response:
We agree that the introductory section can be condensed and more implications can be added in the Abstract.

Changes in manuscript (Page numbers/Line numbers):
We have modified Abstract (P1/L10-11, L25).

**Comment 2**:
"L.15-17: Split in two sentences for clarity."

Response:
Agreed. This has been changed.

Changes in manuscript (Page numbers/Line numbers):
Corrected as suggested (P1/L15-16).

**Comment 3**:
"L.44-47: Shouldn't it be highlighted here the potential/efficiency of adding crushed basalt is somewhat dependent on the basalt composition itself, gran size, soil temperature, moisture and drainage? Furthermore, what are the possible impacts (positive or negative) of ERW for soils themselves and downstream (if/when leachates escape to the surrounding environment)?"

Response:
We fully agree that any pH shift as a result of basalt dissolution will reflect the factors suggested by the reviewer. We have attempted to emphasize this, and also cite papers that discuss potential

impacts of ERW on rivers and oceans (e.g., Zhang et al., 2022; Kanzaki et al., 2023) in lines 38-39.

Changes in manuscript (Page numbers/Line numbers):
We have modified the relevant sentence (P2/L43-44).

**Comment 4**:
"L.48-52: Is there comparative evidence to bridge gaps between pH(s) and pH(pw), as predicted by models?"

Response:
pH(s) and pH(pw) can be relatively close to one another, depending on soil condition and extractants used for soil pH measurements. As far as we know, however, there have been few efforts to mechanistically understand the gap between pH(s) and pH(pw), as we describe in lines 54-56.

Changes in manuscript (Page numbers/Line numbers):
We have added a sentence that further study that reports both porewater pH and soil pH is desirable in Conclusions (P11/L331-333).

**Comment 5**:
"L.81-82: What are the solid species involved in cation exchange?"

Response:
Clay minerals and organic matter are major cation-exchangers in soils. Accordingly clay minerals and organic matter compounds have non-zero CEC values but the rest have zero for CEC in the default setting of the model, although the code allows the user to assign any cation exchange parameterization to any solid species.

Changes in manuscript (Page numbers/Line numbers):
We have added explanations such as those above (P3/L81-82).

**Comment 6**:
"L.130-138: It is probably easier to follow here as a table or as single-line items instead of a running text."

Response:

We agree that a list of symbols used in this study might be useful.

Changes in manuscript (Page numbers/Line numbers):
We have added a list of symbols used in this study in Appendix (P12/L348-349, P26-27).

**Comment 7**:
"L.138-144: This would read best right after Eq.(11), then followed by the table/list of individual parameters (L.130-138)."

Response:
Agreed.

Changes in manuscript (Page numbers/Line numbers):
Corrected as suggested (P5/L131-138).

**Comment 8**:
"L.160-161: Is the pH(s) modelling a two-step process (i.e., a "normal" field run is required to provide the needed boundary conditions for the "lab" experiment.)? Can it be run stand-alone with assumed boundary conditions? Can one obtain only set depths or averaged conditions, or is it possible to calculate the pH(s) continuously along the soil profile?"

Response:
In most cases, field observations cannot be directly utilized as inputs to laboratory experiments. This is because thermodynamic constants for cation exchange are important inputs to the laboratory experiments, but cannot be directly measured in the field and must be determined by developing a model that explains the field observations (e.g., a "field simulation" as described in Section 3). Therefore, we would argue that it makes the most sense to run simulations under field conditions in order to reproduce field observations (obtaining soil physicochemical properties through tuning), and then calculate soil pH in-silico under laboratory conditions.

We attempt to explicitly clarify that "Data from the field run are retrieved at a given model field depth and/or averaged over a given depth interval" in line 160. Accordingly, soil pH can be considered a representative value either at a given depth or over a given depth interval.

Changes in manuscript (Page numbers/Line numbers):
We modified Figure 1 so that the soil pH calculation procedure is clearer (P28).

We have added an explanation that soil pH in Figs. 7 and 8 are calculated at each depth point and not averaged over depth interval to the captions of Figs. 7 and 8 (P34-35).

**Comment 9:**
"L.180-181: Why is DIC added as labile organic matter?"

Response:
This is because of the convenience that adding labile organic matter as a source of DIC is not accompanied by adding any cations. This should be in contrast with the more complicated case where DIC is added as e.g., carbonates, where the amount of cations added as carbonates has to be subtracted from the total amount of cations that are added as oxides/salts to be consistent with the total amount of cations constrained from observations. However, this was also confusing to Reviewer 1, suggesting the need to clarify more fully the reason for this procedure.

Changes in manuscript (Page numbers/Line numbers):
We added an explanation such as above to the relevant sentence (P7/L178-179).

**Comment 10:**
"L.181-190: This would read better as an equation, followed by the list of components."

Response:
In the relevant sentences, we describe the procedure to determine boundary conditions for "laboratory" experiments where soil pH is calculated in multiple steps, and we do not think such multiple-step procedure can be well-formulated as an equation. However, we agree that some sort of list would be useful for readers and have added this to the revised manuscript.

Changes in manuscript (Page numbers/Line numbers):
We added a list of all symbols used in this study in Appendix A (P12/L348-349, P26-27).

**Comment 11:**
"L.208-209: Is such difference between pH(s) and pH(pw) systematic or predictable in any way based on boundary conditions/assumptions?"

Response:
This is a very interesting point. Very little data is currently available for attempting a systematic analysis for the relationship between the two pH measurements. Theoretical prediction might be possible with the current model but there are so many factors that can affect soil pH calculation for a given porewater pH (e.g., thermodynamic parameters for cation exchange, CEC, climate etc.). For instance, the trend found in Miller and Kissel (2010) is among soil pH measurements with different extractants and the study does not include porewater pH data.

Changes in manuscript (Page numbers/Line numbers):
We added a sentence to describe that further comparison of the model with observations will enhance our understanding of the difference between soil pH and porewater pH (P11/L331-333).

**Comment 12**:
"L.219: A brief description here and/or an appendix/SI is needed."

Response:
We agree that a general brief description might be useful to the reader.

Changes in manuscript (Page numbers/Line numbers):
We added a brief description (P8/L225-226).

**Comment 13**:
"L.245-248: Did you test the effect of not including NO3 and Cl to the lab runs?"

Response:
Yes. Without those anions, calculated electrical conductivity is lower and soil pH is higher.

Changes in manuscript (Page numbers/Line numbers):
We mentioned electrical conductivity in the relevant sentence to further justify that those anions exist as salts and should be added as salts in the laboratory runs (P9/L256).

**Comment 14**:
"L.250-251: The model seems to slightly underestimate Ca and Mg. Any particular reason?"

Response:
Slight offsets might have been caused by development of chemical gradients adopting two different CEC values for the two exchangers especially for the relatively strongly bound Ca and Mg.

Changes in manuscript (Page numbers/Line numbers):
We added a sentence such as above (P9/L260-261).

**Comment 15**:

"L.254-256: Can such variability provide any predictability of pH(s) concerning the media used to exchange cations from the solid-phase?"

Response:
This is an interesting question, and we have attempted to address the issue in our response to Comment 11 above. There are many contributing factors that are related to the exchange properties of the solid phase and thus impact prediction of soil pH (e.g., thermodynamic constants for cation exchange, CEC, concentrations of salts, etc.).

Changes in manuscript (Page numbers/Line numbers):
Please see our changes in manuscript in response to Comment 11 by the reviewer.

**Comment 16**:
"L.260-262: 1) The results shown here are based and compared to one mesocosm experiment. Although it looks the model performs well, there is not much to compare in terms of distinct set up conditions. Do these comparisons hold for another mesocosm conditions and natural/agricultural soils?"

Response:
We agree that studies on a variety of soil types are ideal to further validate the framework developed in this study. However, there are unfortunately not so many observations that report comprehensive soil chemistry including detailed porewater chemistry (including pH), soil pH and concentrations of exchangeable cations and thus allow validity tests of the method.

Changes in manuscript (Page numbers/Line numbers):
We have added a sentence mentioning further comparison is desirable in the future study (P11/L331-333).

**Comment 17**:
"2) How does the model closely reproduce pH(s) of previously publish data? This is not shown."

Response:
This was the intention of comparing model simulations to the results from Miller and Kissel (2010) in Fig. 5, which are based on US soils.

Changes in manuscript (Page numbers/Line numbers):
We added the reference to the sentence to avoid confusion (P10/L273).

**Comment 18**:
"L.275: Are these target pH values for the average soil profile or at a specific depth?"

Response:
We clarify that soil pH was calculated as an average over top 15 cm in line 290. On the other hand, soil pH values shown in Figs. 7 and 8 are calculated using bulk soil composition at each depth point. We clarified this difference in the revised manuscript.

Changes in manuscript (Page numbers/Line numbers):
We added more explanations to avoid confusion (P10/L290-291, P11/L311-312, P34-35).

**Comment 19**:
"L.288-289: 1) Would these basalt amendments/pH correction change in any meaningful way under different environmental conditions (e.g., temperature, soil moisture)?"

Response:
Yes, we would expect that the required amount of basalt for a given target pH will change with climate and soil hydrology.

Changes in manuscript (Page numbers/Line numbers):
We modified a sentence to emphasize the importance of climate (P2/L43-45).

**Comment 20**:
"2) Are these predicted basalt contents in line with any expected or suggested plans of soil amendment?"

Response:
The calculated amount is consistent with the range summarized in Swoboda et al. (2022).

Changes in manuscript (Page numbers/Line numbers):
We modified a sentence to indicate that the calculated range is not inconsistent with the amendment levels examined before (P10/L301-P11/L302).

**Comment 21**:
"L.289-290: From Fig. 6-7, it is unclear to me when the target pH are met. Except for target pH(s) = 6.2 (Fig. 6a), after one year all pH values largely exceed the target at the surface. Is that true or

am I miss-interpreting the figure? If pH values would reach > 7.0, what consequences would that bring? Either way, it's not clear. Can you indicate in each panel the target pH (vertical dotter line or so)?"

Response:
We clarify that target pH is met in 1 year in line 287. We agree that interpretation of pH from the figures might not necessarily be straightforward because target pH is evaluated based on average over top 15 cm while the figures show soil pH and porewater pH at each depth points.

Changes in manuscript (Page numbers/Line numbers):
We added more explanations to avoid confusion (P10/L276, 290-291, P11/L311-312, P34-35).
    We added vertical dotted lines to show target pHs in Figs. 7 and 8 (P34-35).

**Comment 22**:
"L.297-298: If so, which approach would be more suitable to adopt when tracking pH, either for natural/agricultural conditions or ERW conditions? Is there a recommendation to be made here, particularly when pH(s) is a more common practice to determine pH of soil systems?"

Response:
We believe that monitoring both porewater pH and soil pH is desirable for mechanistic understanding. However, our point of showing these example simulations is not to make any recommendations but to show the potential difference between porewater pH and soil pH and its impacts on ERW. Typical field measurements will be on soil pH, while many modeling results are reported in terms of porewater pH. The key point for our purposes is to highlight this distinction and emphasize that model-data comparisons need to be mindful of this moving forward.

Changes in manuscript (Page numbers/Line numbers):
We added a sentence that indicates further observations will be desired for further understanding of ERW impacts on croplands (P11/L331-333).

**Comment 23**:
"L.298-300: In these basalt amendment scenarios, are there estimates of other "by products" (e.g., SiOH4 enrichment, secondary precipitation processes) and their fate?"

Response:
As stated in lines 278-280 and might be inferred from the aqueous and solid species tracked in these experiments listed in Table 11, our example experiments here are simplified as much as possible. Therefore, neither Si enrichment nor secondary precipitation is considered.

Changes in manuscript (Page numbers/Line numbers):
We modified the relevant sentence to be clearer (P10/L278-279).

**Comment 24**:
"L.305-310: Can these findings provide any recommendation or guidance for pH measurements in soil and/or ERW practices?"

Response:
We have addressed essentially the same point in response to Comment 22 by the reviewer.

Changes in manuscript (Page numbers/Line numbers):
Please see our change in manuscript in response to Comment 22 by the reviewer.